# B-XAIC Dataset: Benchmarking Explainable AI for Graph Neural Networks Using Chemical Data

## Abstract

Understanding the reasoning behind deep learning model predictions is crucial in cheminformatics and drug discovery, where molecular design determines their properties. However, current evaluation frameworks for Explainable AI (XAI) in this domain often rely on artificial datasets or simplified tasks, employing data-derived metrics that fail to capture the complexity of real-world scenarios and lack a direct link to explanation faithfulness. To address this, we introduce B-XAIC, a novel benchmark constructed from real-world molecular data and diverse tasks with known ground-truth rationales for assigned labels. Through a comprehensive evaluation using B-XAIC, we reveal limitations of existing XAI methods for Graph Neural Networks (GNNs) in the molecular domain. This benchmark provides a valuable resource for gaining deeper insights into the faithfulness of XAI, facilitating the development of more reliable and interpretable models.

## 1 Introduction

Graph Neural Networks (GNNs) have become the standard for predictive modeling of small molecules (Wieder et al., 2020), achieving exceptional performance across property prediction, virtual screening, and related pharmaceutical tasks. While their predictive capabilities are well-established, a growing emphasis is now placed on understanding their reasoning (Jiménez-Luna et al., 2020). In scientific applications of deep learning to small molecules, transparent explanation mechanisms are not merely desirable but crucial. They build researcher trust, ensure model reliability, and potentially uncover novel insights that can accelerate drug discovery (Wu et al., 2023).

To address this gap, a range of Explainable AI (XAI) techniques have been adapted or specifically designed for GNNs, aiming to reveal the mechanisms behind their predictions (Jiménez-Luna et al., 2020; Kakkad et al., 2023). These approaches generally fall into two categories: counterfactual methods(Chen et al., 2022; Lucic et al., 2022; Tan et al., 2022), which seek to identify minimal input changes that alter a model's prediction, and factual methods (Luo et al., 2020; Schlichtkrull et al., 2021; Ying et al., 2019), which aim to highlight important substructures within the input graph. Factual methods further diverge into post-hoc explainers (Ying et al., 2019), that analyze a trained black-box model, and inherently interpretable architectures Feng et al. (2022); Velickovic et al. (2018); Zhang et al. (2022), which aim for transparency through their design. However, recent findings indicate a critical challenge: regardless of the specific XAI method used, the resulting explanations can be unreliable, or even misleading, potentially interfering with scientific understanding Faber et al. (2021). Despite the strong performance of these models on established benchmarks and metrics for GNNs and small molecules, this is still observable.

In response to the limitations of current XAI evaluation for GNNs, the community has developed synthetic datasets (Agarwal et al., 2023; Azzolin et al., 2023; Luo et al., 2020; Wu et al., 2022; Ying et al., 2019). However, these often lack real-world complexity, while creating real-world datasets with ground truth explanations is challenging or impossible. Existing real-world datasets like MUTAG are small and task-limited. Furthermore, many evaluation methods rely on thresholding importance maps or selecting top-k elements, which can be problematic for tasks dependent on the presence/absence of substructures, where no single element is inherently more important. This arbitrary selection can yield inaccurate explanations and misleading metrics. While AUROC (Bajaj et al., 2021; Zhang et al.,

2020) avoids thresholding, it becomes ineffective when no specific element is important, leading to empty ground truth explanations.

To address these limitations, we introduce B-XAIC (Benchmark for eXplainable Artificial Intelligence in Chemistry), a novel benchmark comprising 50K small molecules and 7 diverse tasks, accompanied by both ground truth labels and corresponding explanations, making accuracy-based metrics a directly applicable and reliable evaluation method. B-XAIC tackles the challenges associated with thresholding explanations or selecting the top-k most important elements by considering two distinct scenarios: (1) cases where a specific part of the input graph constitutes the explanation, which can be effectively evaluated using AUROC or Average Precision (AP), and (2) cases where the entire graph is equally important for the prediction, in which the evaluation focuses on ensuring the explanation does not contain irrelevant outliers. Ultimately, B-XAIC enables a direct and fair comparison of various factual XAI approaches, both post-hoc explainers and inherently self-explainable models.

## 2 RELATED WORK

**Explanability in GNNs.** Recent research in Graph Neural Networks (GNNs) has increasingly focused on developing methods to interpret and explain the decisions made by these models. Explainable AI (XAI) techniques for GNNs can be broadly categorized based on the type of explanation they provide (Kakkad et al., 2023). These methods may involve identifying key substructures within the input graph that influence the model's predictions, offering factual explanations by highlighting relevant parts of the input (Dai & Wang, 2021; Luo et al., 2020; Ying et al., 2019; Yuan et al., 2021), or generating counterfactual examples where the input is perturbed in such a way that it leads to a different prediction outcome (Chen et al., 2022; Lucic et al., 2022; Tan et al., 2022).

Furthermore, factual methods for explaining GNN predictions can be broadly classified into post-hoc and self-interpretable approaches. Post-hoc methods aim to explain the predictions of a pre-trained GNN by identifying important nodes, edges, or features that influence the model's decision (Luo et al., 2020; Schlichtkrull et al., 2021; Ying et al., 2019). In contrast, self-interpretable methods design the GNN architecture to inherently incorporate explainability using information constraints, such as attention blocks (Miao et al., 2022; Velickovic et al., 2018) or bottlenecks (Wu et al., 2020), or integrating structural constraints like prototypes (Rymarczyk et al., 2023; Zhang et al., 2022) or graph kernels (Cosmo et al., 2025; Feng et al., 2022), to ensure that the model is more interpretable.

**Explainability Benchmarks.** The need for appropriate datasets to evaluate GNN explainability techniques has led to the introduction of various benchmark datasets with ground-truth explanations. Several synthetic datasets have been developed for node classification and graph classification tasks, where specific motifs serve as the ground truth. For instance, datasets like BA-Shapes, BA-Community, Tree Cycle, and Tree Grids (Ying et al., 2019) are designed for node classification, with the task of predicting whether a node is part of a known motif (such as a cycle, house, or grid). Similarly, synthetic datasets like BA-2Motifs (Luo et al., 2020), BAMultiShapes (Azzolin et al., 2023) and Spurious Motifs (Wu et al., 2022) are designed for graph classification tasks, where the goal is to detect presence of given motifs in the entire graph.

ShapeGGen (Agarwal et al., 2023) is a more recent development in the field of graph benchmarks. It is a synthetic graph generator designed to create a variety of graph datasets with diverse characteristics. While ShapeGGen provides valuable synthetic data, it remains limited by its artificial nature, which may not fully capture the complexity and noise present in real-world datasets.

In addition to synthetic datasets, real-world datasets have been crucial for testing GNN explainability methods. Molecular datasets are particularly valuable in this context, as they can provide ground-truth explanations based on known chemical properties. Examples of such datasets include MUTAG (Debnath et al., 1991), Benzene, Fluoride-Carbonyl, and Alkane-Carbonyl (McCloskey et al., 2019; Sanchez-Lengeling et al., 2020), which are graph classification tasks where explanations are based on the presence or absence of simple chemical structures. These datasets contain 1.8K, 12K, 8.7K and 1.1K graphs, respectively. Although simple, these datasets serve as effective benchmarks and are commonly used in the field. However, experimental datasets like MUTAG, in addition to simple known patterns such as the nitro group, may include more subtle dependencies, making it impossible to assign the absolute ground truth to chemical structures.

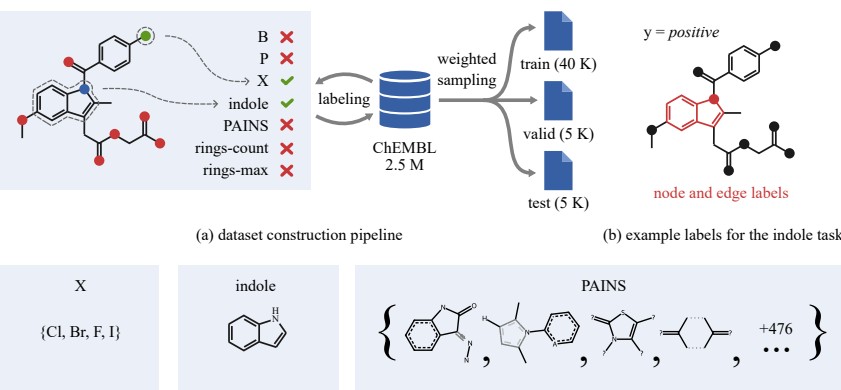

(a) dataset construction pipeline

(b) example labels for the indole task

(c) patterns corresponding to the structure detection tasks

Figure 1: Schematic of our B-XAIC dataset and benchmark; (a) the dataset preparation steps include compound labeling, filtering, and sampling to the training, validation, and testing subsets; (b) for each positive example, atom and bond labels are provided to assess model explanations; (c) the patterns for the halogen and indole tasks are presented, as well as four example PAINS patterns.

Furthermore, many XAI GNN methods are evaluated on more complex molecular datasets where ground-truth explanations are not available, such as NCI1 (Wale & Karypis, 2006), BBBP (Martins et al., 2012), Tox21 (Mayr et al., 2016), or Proteins (Borgwardt et al., 2005). In addition to these, several domain-specific GNN benchmarks are used to assess explainability and related metrics in fields such as visual recognition, natural language processing, or fairness. In the visual domain, datasets like MNIST-75sp (Knyazev et al., 2019) transform images into graphs, where each superpixel is treated as a graph node to test GNN systems. In the textual domain, datasets like Graph SST2, Graph SST5, and Graph Twitter (Yuan et al., 2020)convert sentiment analysis datasets into graph-structured data, allowing for the evaluation of GNN explainability in natural language processing tasks. In the fairness domain, social graph datasets like German Credit, Recidivism, and Credit Defaulter are used to evaluate fairness when dealing with sensitive data (Agarwal et al., 2021). These domain-specific benchmarks provide valuable insights into how GNNs can be interpreted across various applications, thereby expanding the scope and relevance of XAI research. However, since ground-truth explanations are often either unavailable or inherently impossible to define, the evaluation of explainability in these contexts shifts from accuracy-based measures to other aspects of XAI systems (Nauta et al., 2023), such as fidelity (Amara et al., 2022; Longa et al., 2025; Zheng et al., 2024), sparsity (Lucic et al., 2022; Yu et al., 2021), sufficiency, necessity (Chen et al., 2022; Tan et al., 2022), or robustness (Bajaj et al., 2021). These metrics typically evaluate how predicted explanations or predicted classes change in response to alterations in the input graph.

Recognizing the need for more robust evaluation, we developed the B-XAIC benchmark. This resource comprises 50,000 diverse examples with 7 tasks, each paired with ground truth explanations that reflect the intricacies of real-world applications. Our evaluation proceeds in two stages: initially, we determine if a method correctly identifies instances with no significant nodes; subsequently, we examine its accuracy in highlighting the relevant subgraph. Fully covering current challenges in benchmarking XAI methods for GNNs. Although our methodology can be easily extended to materials or proteins, we chose to focus on organic compounds because other domains might need different modeling techniques, which would make benchmarking more challenging.

## 3 BENCHMARK

In this section, firstly we will introduce the B-XAIC dataset that is the core of our benchmark, and then we will provide details on how to evaluate explanations using B-XAIC.

### 3.1 DATASET CONSTRUCTION

Our benchmark dataset was sourced from ChEMBL 35 (Gaulton et al., 2012), which is a public database of 2.5 M molecules with drug-like properties shared under the CC BY-SA 3.0 license.

The molecules were pre-filtered by removing invalid or duplicated SMILES strings. The solvent molecules and counterions were removed to keep only one molecular graph per example.

The benchmark tasks are based on the presence of chemical substructures (see Figure 1), with increasing difficulty of chemical patterns:

(1) Detection of organoboron and organophosphorus compounds. The goal of these two tasks is to predict if a compound contains a boron (B) or phosphorus (P) atom, respectively.

(2) Detection of halogens (X). The prediction should be positive if any of the halogen atoms (bromine, chlorine, fluorine, and iodine) are present in the molecule. This task verifies whether GNNs can find one of multiple alternative patterns in the graph.

(3) Detection of indoles, a bicyclic structure with a benzene ring fused to a pyrrole ring. Compounds containing this structure are widely distributed in nature. In this task, a GNN should effectively pass messages between nodes to detect a larger pattern in the graph.

(4) Detection of pan-assay interference compounds (PAINS). Some chemical structures tend to produce false-positive results in high-throughput screens. We use the list of such patterns proposed by Baell and Holloway (Baell & Holloway, 2010). This task aims to test whether a GNN is able to learn multiple more complex patterns.

(5) Counting rings. The model should predict if a molecule contains more than four rings. This task involves both detecting a pattern and counting its occurrences. Spiro, fused, and bridged rings are considered distinct rings. For example, spiro[5.2]octane will be identified as having two rings.

(6) Detecting large rings with more than six atoms. This task involves counting nodes within a substructure. Spiro, fused, and bridged rings are considered distinct rings.

The final dataset is produced by sampling 50 K molecules from the pre-filtered ChEMBL dataset. Weighted sampling is used to avoid huge class imbalance. The weights are defined as the product of the ratios between the majority and minority classes for all tasks. The average size of a graph in the resulting dataset is 34.56. The data is split randomly into training, validation, and testing sets using the 8-1-1 ratio. The dataset contains a set of binary task labels for each compound and two sets of explanation labels, one for the atoms and one for the edges involved in each detected pattern. Figure 2 illustrates the diversity of the graphs in the dataset.

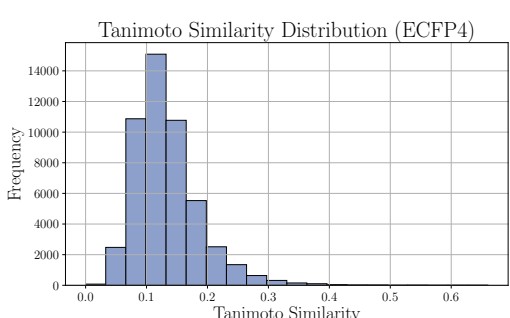

Figure 2: Histogram of Tanimoto similarities illustrating the diversity of the dataset.

## 3.2 Evaluation Metrics

For each task and each graph, the ground truth explanation is defined as a subset of nodes and edges that are relevant to the task.

These explanations fall into two categories:

- **Null explanations (NE)** – where no nodes or edges are more important than others. For example, in task B, if atom B is not present in the graph, then no specific substructure is considered relevant.
- **Subgraph explanations (SE)** – where only a part of the graph is relevant to the task. For instance, in task B, if atom B is present, that node constitutes the explanation.

We evaluate these two groups separately:

- **Null explanations**: The predicted explanation should be uniform for all nodes and edges, without highlighting specific substructures as more important. This requirement can be formally restated as no outliers among the node and edge explanations. To measure this, we use the interquartile range (IQR) method. A prediction is assigned a score of 1 if no outliers are detected, and 0 otherwise.
- **Subgraph explanations**: Because it may be difficult to find the optimal threshold for explanation methods to extract all relevant nodes or edges, we rely on the AUROC metric to test if the most significant nodes and edges are prioritized over the remaining graph structure.

This evaluation is done separately for node-based and edge-based explanations. Table 1 contains relevant details.

**Justification to use IQR.** Our benchmark's NE samples are designed for molecules where no atoms or bonds are important for the task. Therefore, a reliable explanation method should assign uniformly low importance scores to all components. Any significant deviation (i.e., a high IQR) would incorrectly highlight unimportant subgraphs.

We use IQR to quantify this expected uniformity; a low IQR confirms that importance scores are indeed uniformly distributed.

Table 1: Summary of Task and Dataset Statistics. SE denotes relevant subgraphs in positive instances, while NE denotes negative instances without explanations.

| task | % of graphs with positive label | % of graphs with NE | % of nodes in SE | % of edges in SE |
|---|---|---|---|---|
| B | 2.18 | 97.78 | 4.13 ± 2.16 | – |
| P | 12.78 | 86.71 | 4.35 ± 2.44 | – |
| X | 56.46 | 44.13 | 6.05 ± 4.07 | – |
| indole | 36.94 | 63.33 | 31.34 ± 12.15 | 31.49 ± 12.16 |
| PAINS | 32.88 | 67.08 | 34.07 ± 14.22 | 31.37 ± 14.63 |
| rings-count | 30.06 | 1.49 | 64.04 ± 16.03 | 61.85 ± 14.84 |
| rings-max | 5.54 | 1.35 | 50.22 ± 17.44 | 47.12 ± 16.39 |

## 4 RESULTS

### 4.1 EXPERIMENTAL SETUP

**Explainers.** We evaluate a range of explanation methods for graph neural networks, including both gradient-based and mask-based approaches. GNNExplainer (Ying et al., 2019) learns a soft mask over the input graph to identify important substructures, while PGExplainer (Luo et al., 2020) is a parametric version that generalizes across instances. PGMExplainer (Vu & Thai, 2020) uses probabilistic graphical models to capture conditional dependencies in the graph for explanation, and FlowX (Gui et al., 2023) leverages gradients through the message-flow process to estimate feature importance. IntegratedGradients (Sundararajan et al., 2017), Saliency (Simonyan et al., 2014), and Input×Gradient (Shrikumar et al., 2016) are gradient-based methods that attribute importance to input features based on their sensitivity. Deconvolution (Mahendran & Vedaldi, 2016; Shrikumar et al., 2016) and GuidedBackprop (Springenberg et al., 2015) refine backpropagation to highlight relevant features more clearly. GraphMask (Schlichtkrull et al., 2021) uses reinforcement learning to learn sparse binary masks, and ShapleyValueSampling (Štrumbelj & Kononenko, 2010) approximates feature importance via Shapley values. We assess both node- and edge-based explanations depending on the capabilities of each method.

Table 2: F1 scores obtained by three GNN architectures and ProtGNN using these architectures as its backbone; the highest scores are highlighted in bold along with the numbers that are not significantly lower according to the one-sided Wilcoxon test

| | GCN | ProtGNN +GCN | GAT | ProtGNN +GAT | GIN | ProtGNN +GIN |
|---|---|---|---|---|---|---|
| B | **99.94 ± 0.05** | 97.38 ± 1.42 | 99.11 ± 0.19 | 98.52 ± 1.69 | **99.96 ± 0.05** | 96.98 ± 0.69 |
| P | **99.98 ± 0.02** | 95.52 ± 7.98 | **99.97 ± 0.01** | 99.51 ± 0.22 | **99.98 ± 0.03** | 99.77 ± 0.14 |
| X | 99.84 ± 0.05 | 99.68 ± 0.05 | 99.18 ± 0.23 | **99.89 ± 0.05** | **99.94 ± 0.02** | 98.74 ± 0.56 |
| indole | 88.33 ± 1.74 | 73.80 ± 16.62 | 67.30 ± 2.21 | 76.17 ± 2.52 | **98.32 ± 0.36** | 95.76 ± 0.89 |
| PAINS | 79.89 ± 0.47 | 56.34 ± 5.57 | 65.02 ± 1.65 | 54.27 ± 0.64 | **92.90 ± 0.54** | 85.88 ± 4.26 |
| rings-count | 87.27 ± 1.61 | 68.74 ± 8.31 | 82.88 ± 0.59 | 71.04 ± 4.89 | **99.62 ± 0.21** | 83.48 ± 0.69 |
| rings-max | 91.25 ± 1.13 | 91.63 ± 0.19 | 91.03 ± 0.79 | 91.63 ± 0.18 | **92.98 ± 0.84** | 91.63 ± 0.19 |

Table 3: Evaluation of the node and edge explanations for the GIN model; the explainers are grouped into three categories: gradient-based (GB), graph-specific (GS), and perturbation-based (PB); the best score and all scores not significantly lower according to the one-sided Wilcoxon test are bolded.

| Class | Explainer | nodes | | | edges | | |
|---|---|---|---|---|---|---|---|
| | | NE | SE | avg | NE | SE | avg |
| GB | Saliency | 0.51±0.12 | 0.85±0.12 | 0.68 | 0.31±0.15 | 0.67±0.06 | 0.49 |
| | Deconvolution | 0.79±0.10 | 0.81±0.09 | **0.80** | 0.38±0.20 | 0.69±0.04 | 0.53 |
| | InputXGradient | 0.54±0.14 | 0.83±0.14 | 0.68 | 0.37±0.14 | 0.65±0.05 | 0.51 |
| | GuidedBackprop | 0.41±0.12 | **0.87±0.09** | 0.64 | 0.30±0.08 | 0.70±0.10 | 0.50 |
| | IntegratedGradients | 0.39±0.26 | 0.85±0.13 | 0.62 | 0.32±0.18 | 0.70±0.06 | 0.51 |
| GS | GNNExplainer | 0.68±0.12 | 0.67±0.08 | 0.68 | **0.56±0.12** | 0.59±0.03 | 0.58 |
| | GraphMaskExplainer | 0.66±0.03 | 0.66±0.07 | 0.66 | 0.35±0.04 | 0.54±0.01 | 0.45 |
| | PGExplainer | – | – | – | 0.02±0.02 | **0.72±0.07** | 0.37 |
| | PGMExplainer | **0.98±0.02** | 0.76±0.18 | **0.87** | – | – | – |
| | FlowX | 0.72±0.13 | 0.77±0.15 | 0.75 | **0.55±0.07** | 0.64±0.03 | **0.60** |
| PB | ShapleyValueSampling | 0.48±0.23 | 0.83±0.15 | 0.65 | 0.18±0.11 | 0.64±0.03 | 0.41 |

**Models.** We apply explainers to popular graph neural network architectures: GCN (Kipf & Welling, 2017), GAT (Velickovic et al., 2018), GIN (Xu et al., 2019), and ProtGNN (Zhang et al., 2022), a prototype-based, interpretable GNN, instantiated with GCN, GAT, and GIN backbones. Results for GIN are reported here; missing results for GCN, GAT, and ProtGNN variants are in the Appendix.

## 4.2 BENCHMARKING

Our benchmark evaluation incorporates multiple GNN architectures, with classification metrics for the designated tasks presented in Table 2. While all evaluated methods demonstrate strong F1 scores across most tasks, GIN consistently outperforms alternative architectures.

As anticipated, the detection of PAINS patterns emerges as the most challenging, requiring the identification of various alert substructures. Several architectures also exhibit limitations in recognizing indole rings, suggesting insufficient capacity to capture extensive substructures within molecular graphs. The ring-counting task similarly presents difficulties for most models, particularly ProtGNN, which is not capable of highlighting disconnected molecular fragments Elhadri et al. (2025).

Given our benchmark's primary focus on comparing XAI methods, subsequent analysis will emphasize results from the GIN architecture, which achieves near-perfect performance across our synthetic tasks. This exceptional performance suggests that GIN formulates predictions based on appropriate chemical principles, making it an ideal candidate for our explainability evaluations.

**Node explanations.** First, we will focus on node explanations. Table 3 shows the results of various XAI methods applied to GIN, our best-performing model. The reported evaluation metrics are averaged across all tasks, and all best results that are not statistically significantly worse than the highest number (according to a one-sided Wilcoxon test) are highlighted in bold. Gradient-based methods are, on average, better at localizing important patterns than other methods. However, they tend to highlight molecular fragments even when the pattern is absent, resulting in low NE scores.

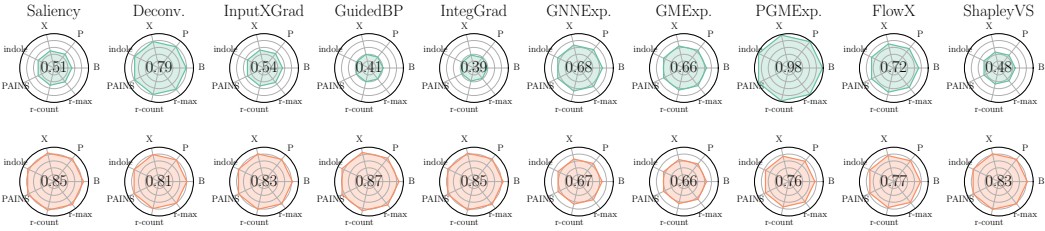

Figure 3: Evaluation of node-level explanations for GIN. Null explanation results are shown in green, and subgraph explanation in orange. Average scores for each method are displayed in the center.

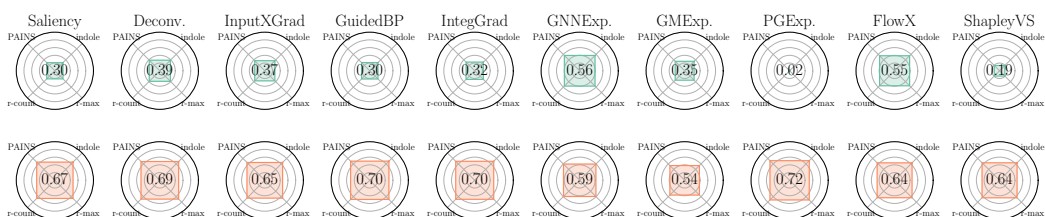

Figure 4: Evaluation of edge-level explanations for GIN. Null explanation results are shown in green, and subgraph explanation in orange. Average scores for each method are displayed in the center.

The detailed results for each task are shown in Figure 3 and further detailed in the Appendix. Two groups of explanation methods are formed. The first one achieves good NE scores and obtains lower SE scores, and the other group exhibits opposite behavior. This suggests that some methods provide more contrastive and precise explanations, while other methods return more uniform attributions. The tasks of finding boron, phosphorus, and halogen atoms are the easiest for all the methods. The most difficult patterns to find are those related to rings, either counting them or measuring their size.

**Edge explanations.**    Similar analysis was conducted for edges, and the results are shown in Table 3. In this case, gradient-based methods do not always outperform the other methods in terms of the SE metric. The strong performance of the methods based on subgraph extraction may be caused by the fact that edges are included in the extracted subgraphs, while gradient methods often focus more on nodes. Interestingly, the GNNExplainer model significantly outperforms all other methods in the NE metric, while GuidedBackprop is best at detecting important edges (one-sided Wilcoxon test).

The results of all explainers are presented in Figure 4, and the exact numbers can be found in the Appendix. In these results, we focus on four tasks that involve edges, excluding all tasks aiming at detecting single atoms. Also in the case of edges, tasks related to counting rings or atoms in rings appear to be more challenging for the explainers.

**Relationship between model performance and explanation quality.**    As illustrated in Figure 9, there exists a notable correlation between model performance and the quality of the explanations generated for prediction outcomes. This relationship is particularly evident in more complex tasks, such as PAINS detection, where SE scores are correlated with F1 scores. The data suggests that models achieving superior predictive accuracy also tend to produce more meaningful structural explanations. In contrast, the correlation between model performance and NE scores appears considerably weaker.

**Explanation examples**    Figure 6 illustrates representative explanations for both positive and negative graph instances. We observe that some techniques tend to highlight atoms proximal to the relevant subgraph, potentially due to limited control over the message-passing mechanism in GNNs. Additionally, we observe that even methodologically similar explanation approaches can generate markedly divergent explanations for the same graph. For the negative instances, there is no universal threshold that can be used across all methods to separate important nodes because one method can attribute weights near zero uniformly for all the nodes, while another method predicts uniform values around 4.5. In both cases no subgraph can be highlighted as predicted to be more significant. All these observations lead to the conclusion that widely used explainers struggle to highlight even simple patterns for GNNs that achieve almost perfect accuracy. This emphasizes the immense need for benchmarks like B-XAIC to accelerate research on new XAI methods for graphs.

### 4.3 DISCUSSION

Our findings clearly demonstrate the critical need for new XAI benchmarks specifically tailored to molecular graphs. Current XAI techniques exhibit significant deficiencies in generating adequate explanations, even for the most elementary tasks proposed in our B-XAIC benchmark. Despite the GIN model achieving remarkably high performance metrics, with F1 scores exceeding 98% for all

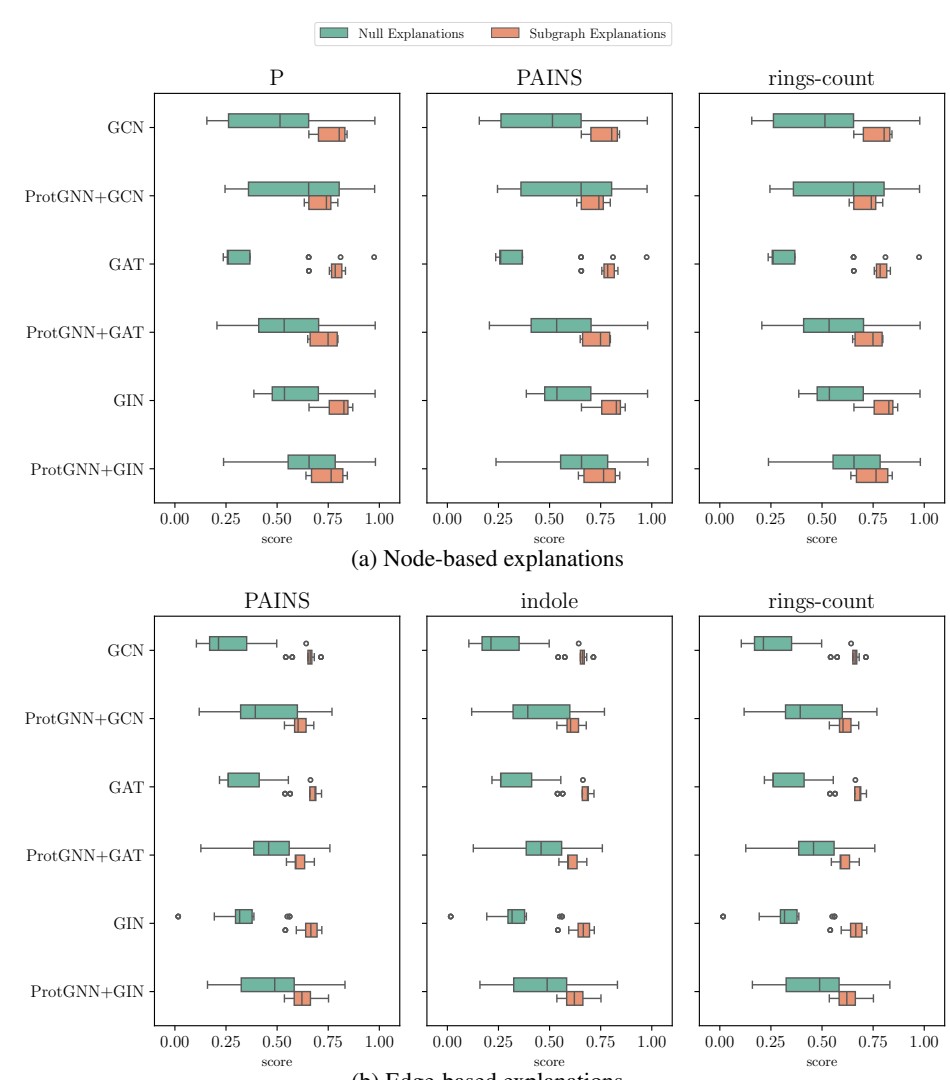

Figure 5: Boxplots showing the distribution of explanation quality across different explainers for each model. Results are aggregated per model, highlighting that some models are inherently more difficult to explain than others.

proposed tasks, the explanations generated by the explainers consistently fail to properly identify and highlight the relevant molecular structures.

While complex nonlinear interactions between atoms undoubtedly characterize real-world chemical applications, our benchmark reveals that incorrect atom attribution persists even in comparatively simple tasks. This systematic failure likely stems from the fundamental architectural principles underlying GNNs. The iterative message-passing layers inherent to these networks result in information diffusion among neighboring nodes, substantially impeding precise localization of salient features. This phenomenon represents a significant challenge to the field and warrants focused investigation into novel approaches that can maintain predictive power while enhancing interpretability.

**Limitations.** The primary limitation of this study lies in its exclusive focus on local explanations. This design choice is justified by the current landscape of GNN explainability methods, where support for global explanations remains limited, hindering a direct and fair comparison across diverse techniques. Furthermore, the utilization of real-world molecular data, while providing real-world data complexity, introduces a potential confound. Despite conducting lots of out-of-distribution experiments (see Appendix), we cannot guarantee that the trained models base their predictions

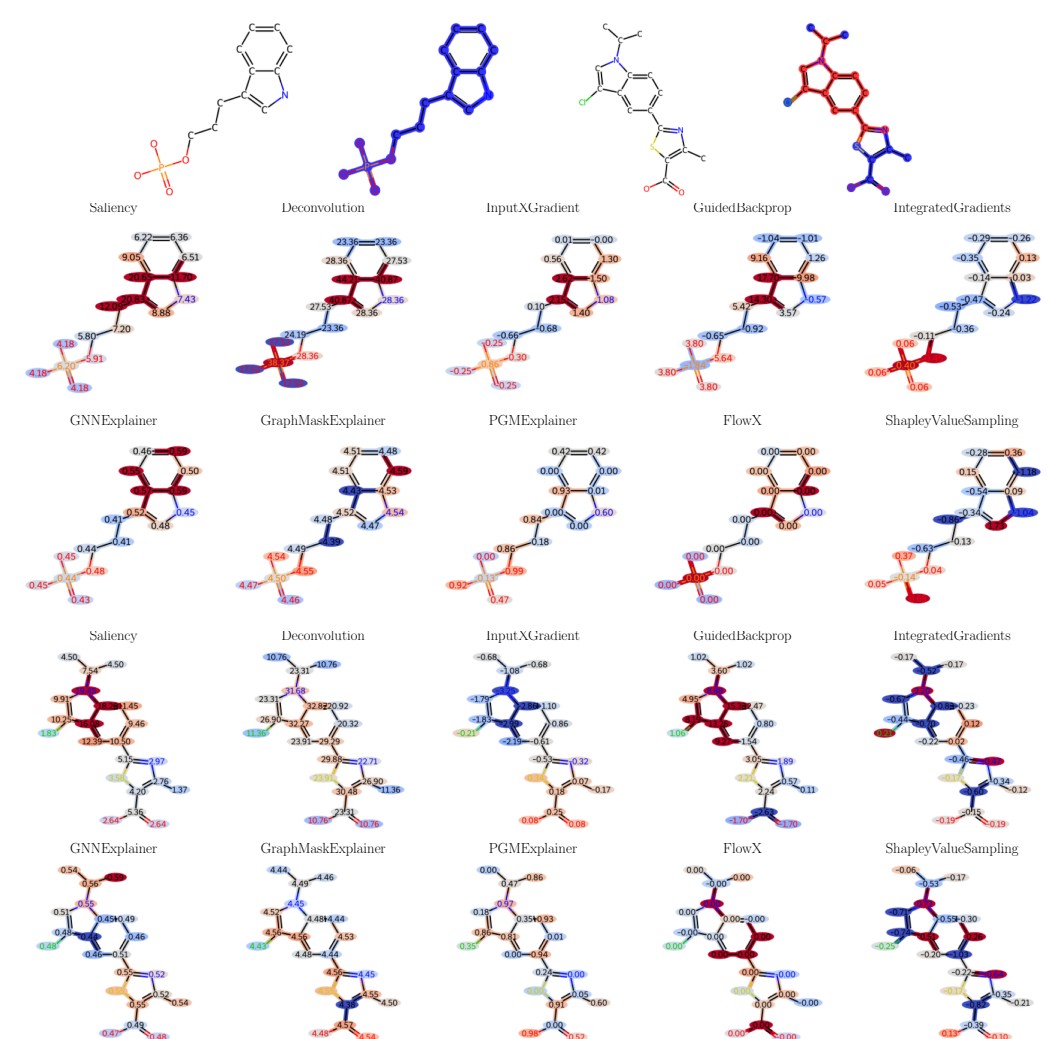

Figure 6: Node-level explanation examples on graphs from different classes in the PAINS task, using the GIN model and different explanation methods.

on the intended underlying chemical principles. Consequently, suboptimal performance of an XAI method on B-XAIC could be attributed to either deficiencies in the explanation technique itself or from the model's failure to learn the task based on the expected structural features. Therefore, a holistic evaluation of explainer performance, considering both explanation quality metrics and the model's predictive accuracy on the test set, is crucial when utilizing this benchmark.

## 5 CONCLUSIONS

In conclusion, the B-XAIC dataset offers a valuable new resource for the GNN XAI community. By providing real-world molecular data with structure-derived ground truth explanations, our dataset enables rigorous benchmarking of both inherently interpretable graph models and post-hoc GNN explainers. The introduction of null and subgraph explanation concepts, along with the edge-based and node-based variations, offers a more nuanced evaluation of XAI capabilities across different explanation types and graph aspects. We believe B-XAIC will serve as a crucial baseline for future research, clearly highlighting the strengths and limitations of emerging XAI methods.

Our ongoing work aims to further enrich this dataset by incorporating activity-cliff scenarios, pushing the boundaries of XAI techniques to uncover subtle but critical distinctions within graph data.

**Ethics Statement.** This work contributes to the broader field of explainable AI (XAI), specifically within the context of graph neural networks applied to drug discovery and molecular modeling. The B-XAIC dataset offers the community a standardized benchmark for evaluating novel XAI techniques dedicated to small molecules. Beyond this specific domain, we anticipate its utility for assessing XAI methods on graphs of moderate size (up to 60 nodes), a common scale in various real-world applications. More generally, this research provides a valuable example for the broader XAI community, demonstrating how real-world data and carefully designed tasks of increasing complexity can be leveraged for effective and insightful XAI benchmarking. Ultimately, we envision that B-XAIC will facilitate the development of more robust and transparent XAI methods for graph data. This advancement holds a promise to enhance the interpretability and trustworthiness of GNNs, allowing for their wider adoption in critical production environments, especially in scientific discovery and the design of new therapeutics.

**Reproducibility Statement.** To ensure accessibility and encourage community engagement, we have hosted the B-XAIC dataset on Hugging Face and provided open-source code for its execution. Furthermore, the careful design of our molecule selection process and data hosting infrastructure allows us to effectively mitigate the risk of data misuse. Importantly, the dataset is released under the CC-BY-SA license, empowering the community to leverage this resource while ensuring proper attribution and continued sharing. You can find the data and code under the following link: `https://anonymous.4open.science/r/B-XAIC-04DE`. We conduct our experiments using an NVIDIA H100 GPU with 80GB of HBM3 memory.

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

## A APPENDIX

Here we provide the full set of results that were shown partially in the main paper. These results accompany the introduction of our benchmark and offer a detailed view of node and edge explanation performance across different explainer types.

### A.1 RANKING BY EVALUATION SCORES

Table 4 and Table 5 report the evaluation metrics for all model–explainer combinations, averaged across tasks and sorted by the aggregated scores for node and edge explanations, respectively.

### A.2 VISUAL SUMMARY OF EVALUATION SCORES

The radar plots in Figure 7 and Figure 8 illustrate the evaluation scores of each model–explainer combination across all 7 tasks, providing a visual comparison of their performance.

Table 4: Ranking of all model–explainer combinations based on the evaluation of **node explanations**, sorted by overall scores. The best score and all scores not significantly lower (according to a one-sided Wilcoxon test) are highlighted in bold. For NE and SE, we report the mean and standard deviation; for the overall score (avg), we additionally provide the standard error of the mean (SEM) to highlight the trade-off between NE and SE performance.

| Model | Explainer | NE | SE | avg |
|---|---|---|---|---|
| ProtGNN+GAT | PGMExplainer | **0.98±0.01** | 0.79±0.20 | **0.89±0.10** |
| GIN | PGMExplainer | **0.98±0.02** | 0.76±0.18 | **0.87±0.10** |
| ProtGNN+GCN | PGMExplainer | **0.98±0.02** | 0.74±0.18 | **0.86±0.09** |
| ProtGNN+GIN | PGMExplainer | **0.98±0.02** | 0.73±0.18 | **0.86±0.09** |
| GCN | PGMExplainer | **0.98±0.02** | 0.73±0.14 | **0.86±0.07** |
| GAT | PGMExplainer | **0.97±0.01** | 0.73±0.18 | **0.85±0.09** |
| ProtGNN+GIN | FlowX | 0.84±0.14 | 0.83±0.11 | **0.84±0.09** |
| ProtGNN+GAT | FlowX | 0.82±0.14 | 0.80±0.08 | 0.81±0.08 |
| GIN | Deconvolution | 0.79±0.10 | 0.81±0.09 | 0.80±0.07 |
| GAT | FlowX | 0.79±0.13 | 0.80±0.12 | 0.79±0.09 |
| ProtGNN+GCN | Deconvolution | 0.80±0.11 | 0.77±0.16 | 0.79±0.10 |
| ProtGNN+GIN | Deconvolution | 0.78±0.13 | 0.79±0.11 | 0.79±0.10 |
| GCN | FlowX | 0.76±0.14 | 0.80±0.11 | 0.78±0.08 |
| GCN | Deconvolution | 0.78±0.11 | 0.77±0.16 | 0.78±0.10 |
| ProtGNN+GCN | GuidedBackprop | 0.90±0.16 | 0.64±0.11 | 0.77±0.11 |
| ProtGNN+GCN | FlowX | 0.76±0.16 | 0.75±0.07 | 0.75±0.09 |
| GIN | FlowX | 0.72±0.14 | 0.77±0.12 | 0.75±0.10 |
| ProtGNN+GIN | Saliency | 0.63±0.21 | **0.84±0.10** | 0.74±0.12 |
| ProtGNN+GIN | GuidedBackprop | 0.80±0.26 | 0.67±0.13 | 0.74±0.12 |
| ProtGNN+GAT | GuidedBackprop | 0.74±0.33 | 0.66±0.15 | 0.70±0.21 |
| ProtGNN+GCN | GNNExplainer | 0.77±0.17 | 0.63±0.10 | 0.70±0.06 |
| ProtGNN+GIN | GNNExplainer | 0.74±0.18 | 0.64±0.11 | 0.69±0.07 |
| GIN | Saliency | 0.51±0.12 | **0.85±0.12** | 0.68±0.10 |
| GIN | InputXGradient | 0.54±0.14 | 0.83±0.14 | 0.68±0.10 |
| GIN | GNNExplainer | 0.68±0.12 | 0.67±0.08 | 0.68±0.06 |
| GCN | GuidedBackprop | 0.51±0.22 | **0.84±0.13** | 0.68±0.10 |
| ProtGNN+GCN | Saliency | 0.62±0.23 | 0.74±0.14 | 0.68±0.11 |
| ProtGNN+GIN | InputXGradient | 0.55±0.27 | 0.80±0.13 | 0.68±0.14 |
| ProtGNN+GAT | GNNExplainer | 0.70±0.18 | 0.65±0.09 | 0.68±0.10 |
| ProtGNN+GAT | Saliency | 0.53±0.35 | 0.80±0.18 | 0.67±0.19 |
| GIN | GraphMaskExplainer | 0.66±0.03 | 0.66±0.07 | 0.66±0.03 |
| ProtGNN+GIN | GraphMaskExplainer | 0.66±0.03 | 0.66±0.07 | 0.66±0.03 |
| GCN | GraphMaskExplainer | 0.65±0.04 | 0.66±0.07 | 0.65±0.03 |
| ProtGNN+GCN | GraphMaskExplainer | 0.65±0.04 | 0.66±0.07 | 0.65±0.03 |
| GAT | GraphMaskExplainer | 0.65±0.04 | 0.66±0.07 | 0.65±0.03 |
| ProtGNN+GAT | GraphMaskExplainer | 0.65±0.04 | 0.66±0.07 | 0.65±0.03 |
| GIN | ShapleyValueSampling | 0.48±0.23 | **0.83±0.15** | 0.65±0.14 |
| ProtGNN+GAT | InputXGradient | 0.49±0.38 | 0.79±0.18 | 0.64±0.21 |
| GIN | GuidedBackprop | 0.41±0.12 | **0.87±0.09** | 0.64±0.09 |
| GCN | Saliency | 0.43±0.13 | 0.84±0.14 | 0.63±0.08 |
| ProtGNN+GIN | IntegratedGradients | 0.44±0.24 | 0.82±0.12 | 0.63±0.13 |
| GIN | IntegratedGradients | 0.39±0.26 | **0.85±0.13** | 0.62±0.16 |
| ProtGNN+GAT | IntegratedGradients | 0.41±0.36 | 0.79±0.18 | 0.60±0.20 |
| GCN | GNNExplainer | 0.53±0.17 | 0.67±0.08 | 0.60±0.09 |
| ProtGNN+GCN | InputXGradient | 0.36±0.32 | 0.76±0.17 | 0.56±0.14 |
| GAT | GNNExplainer | 0.37±0.17 | 0.76±0.16 | 0.56±0.13 |
| GAT | InputXGradient | 0.26±0.22 | 0.83±0.15 | 0.55±0.16 |
| GAT | ShapleyValueSampling | 0.27±0.20 | 0.82±0.15 | 0.54±0.15 |
| ProtGNN+GCN | IntegratedGradients | 0.29±0.27 | 0.80±0.15 | 0.54±0.14 |
| GAT | IntegratedGradients | 0.25±0.20 | 0.82±0.15 | 0.54±0.15 |
| GCN | ShapleyValueSampling | 0.26±0.15 | 0.80±0.17 | 0.53±0.10 |
| GCN | InputXGradient | 0.21±0.19 | 0.83±0.16 | 0.52±0.10 |
| GAT | GuidedBackprop | 0.26±0.23 | 0.78±0.18 | 0.52±0.17 |
| GAT | Deconvolution | 0.26±0.23 | 0.78±0.18 | 0.52±0.17 |
| GAT | Saliency | 0.24±0.22 | 0.79±0.18 | 0.51±0.17 |
| ProtGNN+GAT | Deconvolution | 0.26±0.20 | 0.74±0.14 | 0.50±0.10 |
| ProtGNN+GIN | ShapleyValueSampling | 0.24±0.24 | 0.74±0.14 | 0.49±0.14 |
| ProtGNN+GCN | ShapleyValueSampling | 0.25±0.27 | 0.73±0.15 | 0.49±0.14 |
| GCN | IntegratedGradients | 0.16±0.13 | 0.82±0.16 | 0.49±0.07 |
| ProtGNN+GAT | ShapleyValueSampling | 0.21±0.31 | 0.75±0.18 | 0.48±0.17 |

Table 5: Ranking of all model–explainer combinations based on the evaluation of **edge explanations**, sorted by overall scores, following the format of Table 4.

| Model | Explainer | NE | SE | avg |
|---|---|---|---|---|
| ProtGNN+GIN | GuidedBackprop | **0.86±0.16** | 0.60±0.05 | **0.73±0.08** |
| ProtGNN+GIN | FlowX | 0.71±0.14 | 0.68±0.01 | **0.69±0.07** |
| ProtGNN+GCN | GuidedBackprop | **0.80±0.22** | 0.58±0.07 | **0.69±0.12** |
| ProtGNN+GAT | GuidedBackprop | **0.79±0.22** | 0.59±0.05 | **0.69±0.11** |
| ProtGNN+GIN | GNNExplainer | 0.76±0.23 | 0.55±0.04 | 0.65±0.11 |
| ProtGNN+GAT | FlowX | 0.63±0.07 | 0.68±0.03 | 0.65±0.04 |
| GAT | FlowX | 0.63±0.08 | 0.66±0.02 | 0.65±0.05 |
| ProtGNN+GIN | Saliency | 0.53±0.28 | **0.76±0.11** | 0.64±0.16 |
| ProtGNN+GCN | FlowX | 0.59±0.07 | 0.68±0.04 | 0.64±0.04 |
| GCN | FlowX | 0.60±0.11 | 0.68±0.04 | 0.64±0.06 |
| ProtGNN+GIN | InputXGradient | 0.55±0.29 | 0.66±0.05 | 0.61±0.15 |
| ProtGNN+GCN | GNNExplainer | 0.64±0.24 | 0.58±0.05 | 0.61±0.12 |
| GIN | FlowX | 0.55±0.07 | 0.64±0.03 | 0.60±0.04 |
| ProtGNN+GAT | GNNExplainer | 0.61±0.23 | 0.57±0.04 | 0.59±0.11 |
| GIN | GNNExplainer | 0.57±0.11 | 0.59±0.03 | 0.58±0.05 |
| ProtGNN+GIN | IntegratedGradients | 0.45±0.28 | 0.70±0.07 | 0.57±0.16 |
| ProtGNN+GAT | PGExplainer | 0.52±0.33 | 0.62±0.10 | 0.57±0.15 |
| ProtGNN+GAT | Saliency | 0.49±0.31 | 0.64±0.10 | 0.56±0.14 |
| GAT | GNNExplainer | 0.55±0.05 | 0.56±0.01 | 0.56±0.02 |
| ProtGNN+GCN | Deconvolution | 0.43±0.21 | 0.68±0.03 | 0.55±0.10 |
| ProtGNN+GAT | GraphMaskExplainer | 0.53±0.05 | 0.55±0.00 | 0.54±0.03 |
| ProtGNN+GCN | GraphMaskExplainer | 0.54±0.05 | 0.53±0.01 | 0.54±0.03 |
| GIN | Deconvolution | 0.38±0.14 | 0.69±0.04 | 0.53±0.07 |
| GCN | GraphMaskExplainer | 0.51±0.07 | 0.54±0.00 | 0.53±0.03 |
| ProtGNN+GAT | InputXGradient | 0.45±0.34 | 0.59±0.06 | 0.52±0.16 |
| ProtGNN+GCN | Saliency | 0.38±0.32 | 0.65±0.10 | 0.52±0.16 |
| GAT | Saliency | 0.36±0.30 | 0.67±0.03 | 0.51±0.15 |
| GIN | InputXGradient | 0.37±0.16 | 0.65±0.05 | 0.51±0.08 |
| GAT | GraphMaskExplainer | 0.48±0.04 | 0.54±0.00 | 0.51±0.02 |
| GIN | IntegratedGradients | 0.32±0.18 | 0.69±0.06 | 0.51±0.10 |
| GIN | GuidedBackprop | 0.29±0.08 | 0.72±0.09 | 0.50±0.05 |
| GAT | InputXGradient | 0.30±0.25 | 0.68±0.03 | 0.49±0.12 |
| GCN | Deconvolution | 0.32±0.23 | 0.67±0.02 | 0.49±0.12 |
| ProtGNN+GAT | Deconvolution | 0.34±0.28 | 0.65±0.04 | 0.49±0.14 |
| GAT | ShapleyValueSampling | 0.31±0.24 | 0.66±0.03 | 0.49±0.12 |
| ProtGNN+GIN | Deconvolution | 0.34±0.17 | 0.63±0.04 | 0.48±0.09 |
| GCN | GuidedBackprop | 0.30±0.26 | 0.66±0.03 | 0.48±0.13 |
| ProtGNN+GCN | InputXGradient | 0.35±0.29 | 0.61±0.07 | 0.48±0.14 |
| GAT | Deconvolution | 0.26±0.22 | 0.69±0.02 | 0.48±0.12 |
| GAT | IntegratedGradients | 0.26±0.22 | 0.69±0.02 | 0.47±0.12 |
| GAT | GuidedBackprop | 0.25±0.23 | 0.69±0.02 | 0.47±0.12 |
| ProtGNN+GCN | PGExplainer | 0.30±0.35 | 0.63±0.08 | 0.47±0.17 |
| GIN | Saliency | 0.26±0.15 | 0.66±0.06 | 0.46±0.07 |
| GCN | GNNExplainer | 0.35±0.23 | 0.57±0.02 | 0.46±0.12 |
| GAT | PGExplainer | 0.18±0.21 | 0.72±0.03 | 0.45±0.11 |
| GIN | GraphMaskExplainer | 0.35±0.04 | 0.54±0.01 | 0.45±0.02 |
| ProtGNN+GIN | GraphMaskExplainer | 0.35±0.04 | 0.54±0.01 | 0.44±0.02 |
| GCN | IntegratedGradients | 0.20±0.16 | 0.66±0.03 | 0.43±0.08 |
| GCN | InputXGradient | 0.17±0.21 | 0.69±0.04 | 0.43±0.11 |
| GIN | ShapleyValueSampling | 0.20±0.09 | 0.63±0.03 | 0.42±0.04 |
| ProtGNN+GCN | IntegratedGradients | 0.19±0.23 | 0.64±0.06 | 0.42±0.12 |
| GCN | PGExplainer | 0.12±0.10 | 0.71±0.04 | 0.41±0.05 |
| GCN | Saliency | 0.16±0.14 | 0.65±0.03 | 0.40±0.07 |
| GCN | ShapleyValueSampling | 0.15±0.17 | 0.65±0.03 | 0.40±0.08 |
| ProtGNN+GIN | ShapleyValueSampling | 0.17±0.28 | 0.58±0.02 | 0.38±0.14 |
| ProtGNN+GIN | PGExplainer | 0.13±0.18 | 0.62±0.06 | 0.38±0.08 |
| ProtGNN+GAT | ShapleyValueSampling | 0.16±0.28 | 0.59±0.04 | 0.38±0.13 |
| ProtGNN+GAT | IntegratedGradients | 0.13±0.28 | 0.61±0.05 | 0.37±0.13 |
| GIN | PGExplainer | 0.02±0.02 | 0.71±0.06 | 0.36±0.03 |
| ProtGNN+GCN | ShapleyValueSampling | 0.13±0.24 | 0.59±0.04 | 0.36±0.12 |

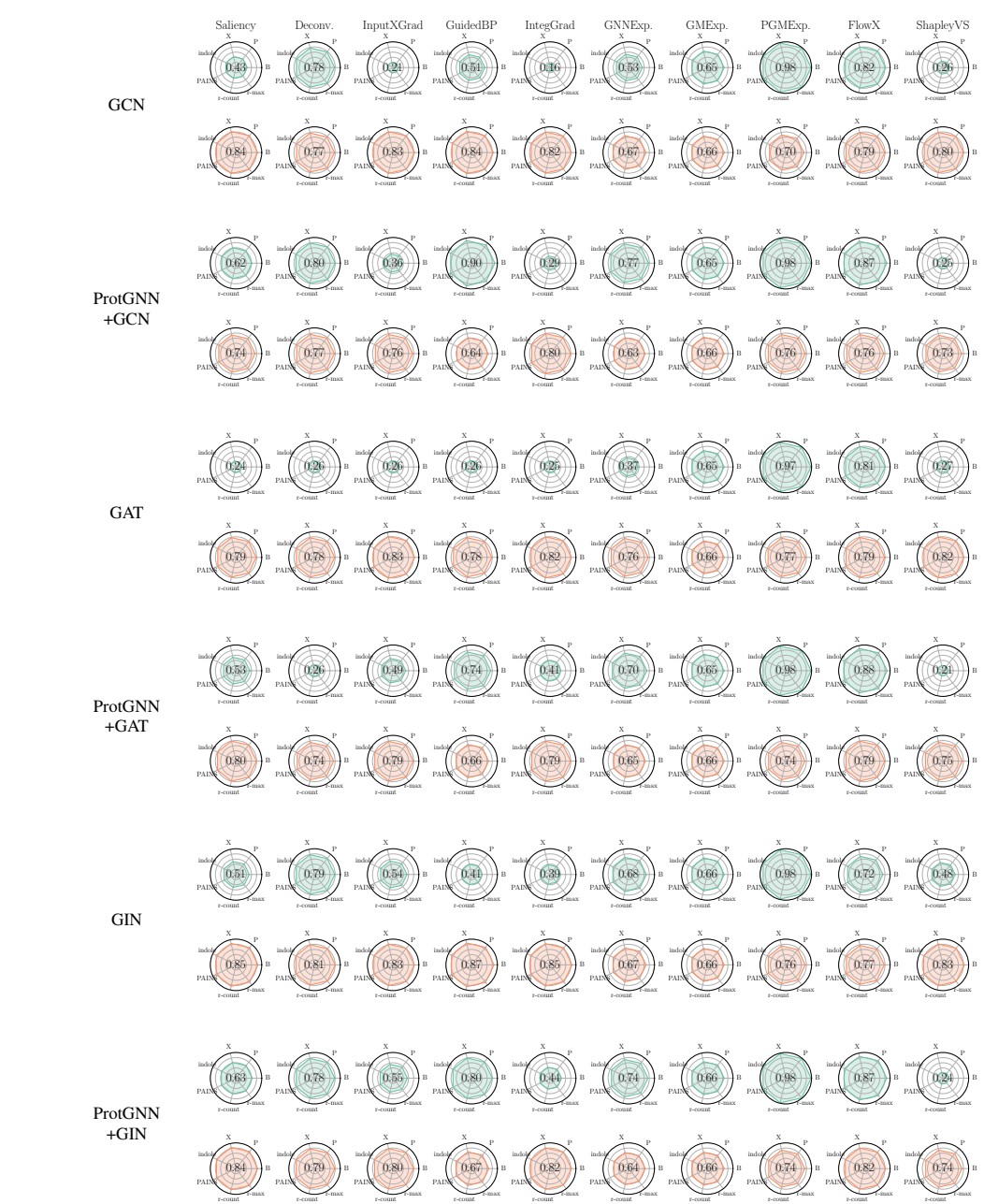

Figure 7: Evaluation of **node explanations** for all model-explainer combinations. Null explanation results are shown in green, and subgraph explanation results in orange. Overall average scores for each method are displayed in the center.

### A.3 VISUALIZATION OF CORRELATION BETWEEN EVALUATION METRICS AND MODEL PERFORMANCE

Figure 9 compare NE and SE metrics for each evaluated model and illustrate the correlation between evaluation scores and model performance for each task.

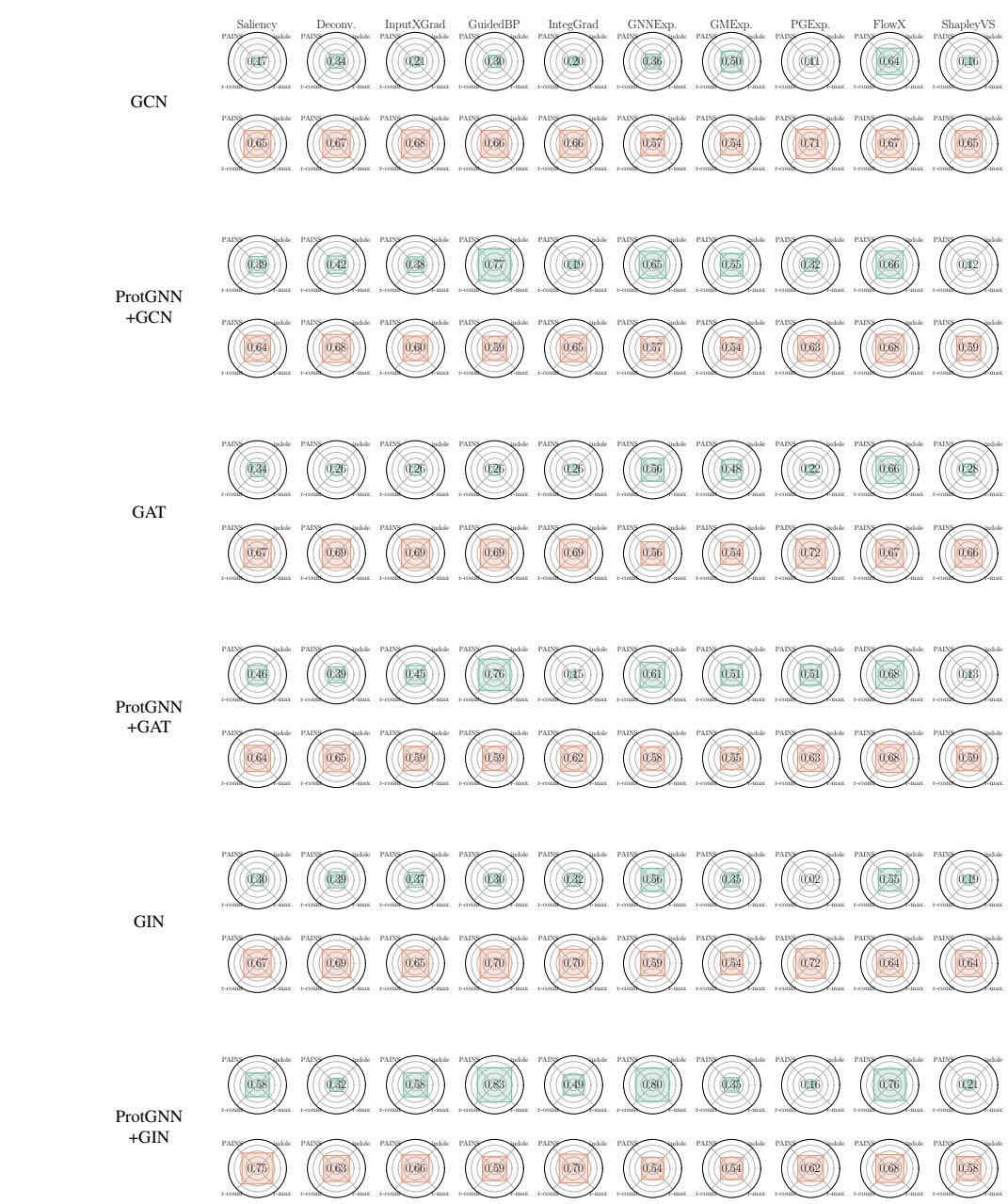

Figure 8: Evaluation of **edge explanations** for all model-explainer combinations. Null explanation results are shown in green, and subgraph explanation results in orange. Overall average scores for each method are displayed in the center.

## A.4 DISENTANGLING MODEL CAPACITY FROM EXPLAINER EXPRESSIVENESS

We conducted experiments evaluating XAI methods in both SE and NE regimes, but only on instances where the GNN model made correct predictions. This approach rigorously separates the XAI method's performance from the backbone model's predictive capabilities.

As illustrated in our Node and Edge explanation tables (Table 7 and Table 6, XAI method performance showed remarkable consistency whether evaluated on all predictions or solely on correct ones (pred=target).

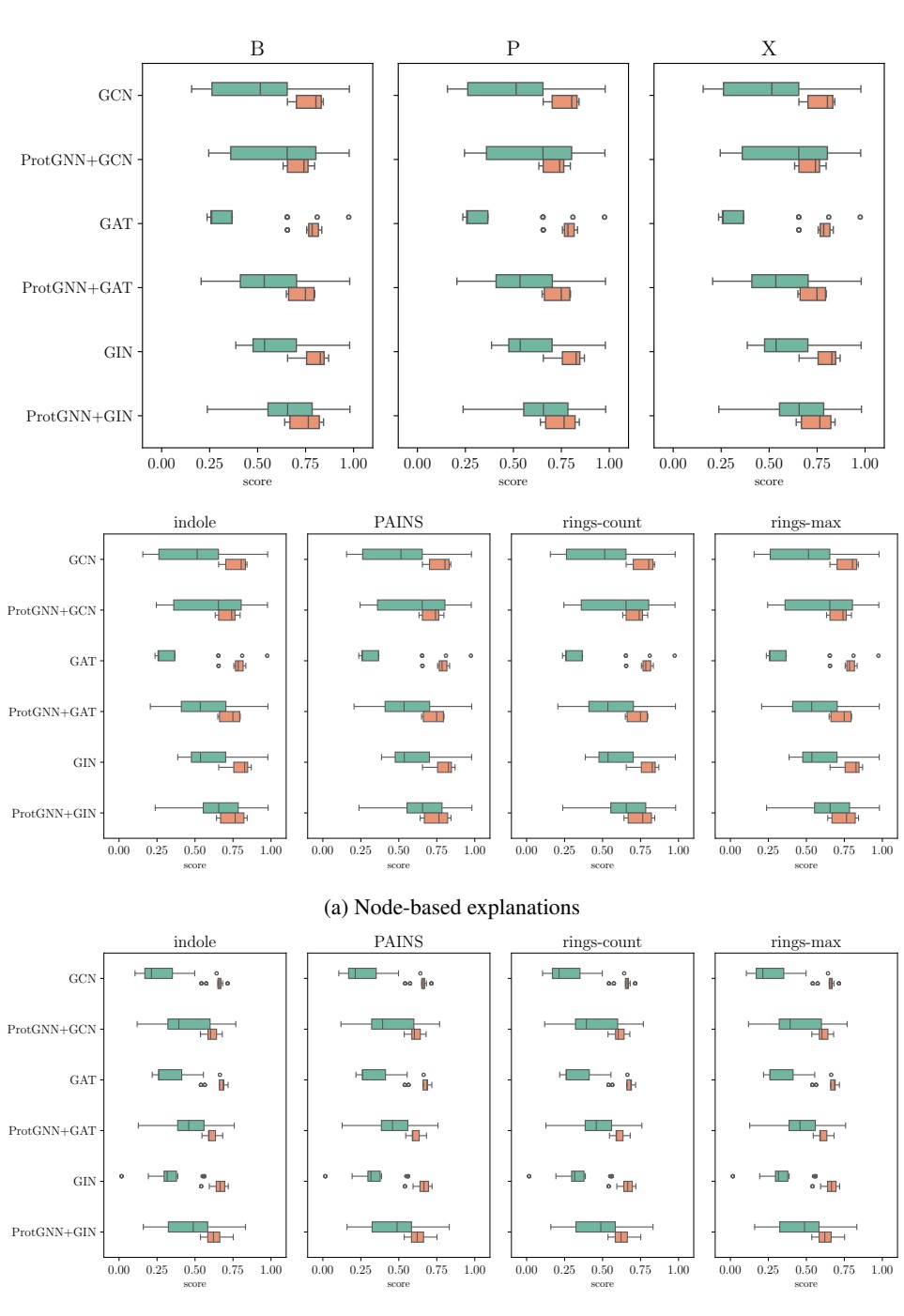

(a) Node-based explanations

(b) Edge-based explanations

Figure 9: Boxplots showing the distribution of explanation quality across different explainers for each model. Results are aggregated per model, highlighting that some models are inherently more difficult to explain than others.

Additionally, our GNN models consistently achieved high accuracy across all tasks. This high baseline performance confirms that the models possess sufficient capacity to learn the underlying chemical principles. Therefore, observed variations in XAI effectiveness can be attributed to the XAI technique itself, rather than limitations in the GNN's ability to learn the task.

Table 6: Node explanation performance only for correct prediction from GIN.

| Task | Type | Subset | Saliency | Deconv. | InputXGrad | GuidedBP | IntegGrad | GNNExp. | GMExp. | PGMExp. | FlowX | ShapleyVS |
|---|---|---|---|---|---|---|---|---|---|---|---|---|
| PAINS | NE | all | 0.39 | 0.81 | 0.44 | 0.32 | 0.07 | 0.68 | 0.64 | 0.99 | 0.7 | 0.21 |
| | | pred=target | 0.42 | 0.81 | 0.44 | 0.32 | 0.07 | 0.68 | 0.64 | 0.99 | 0.72 | 0.21 |
| | SE | all | 0.79 | 0.44 | 0.72 | 0.79 | 0.69 | 0.63 | 0.60 | 0.62 | 0.64 | 0.64 |
| | | pred=target | 0.80 | 0.70 | 0.72 | 0.80 | 0.69 | 0.63 | 0.60 | 0.62 | 0.64 | 0.64 |
| rings-max | NE | all | 0.48 | 0.83 | 0.55 | 0.40 | 0.38 | 0.67 | 0.65 | 0.98 | 0.64 | 0.60 |
| | | pred=target | 0.48 | 0.83 | 0.54 | 0.42 | 0.44 | 0.67 | 0.65 | 0.98 | 0.64 | 0.59 |
| | SE | all | 0.70 | 0.77 | 0.67 | 0.75 | 0.67 | 0.61 | 0.58 | 0.59 | 0.65 | 0.63 |
| | | pred=target | 0.70 | 0.79 | 0.66 | 0.75 | 0.67 | 0.62 | 0.57 | 0.57 | 0.65 | 0.65 |

Table 7: Edge explanation performance only for correct prediction from GIN.

| Task | Type | Subset | Saliency | Deconv. | InputXGrad | GuidedBP | IntegGrad | GNNExp. | GMExp. | PGExp. | FlowX | ShapleyVS |
|---|---|---|---|---|---|---|---|---|---|---|---|---|
| PAINS | NE | all | 0.17 | 0.39 | 0.23 | 0.28 | 0.19 | 0.56 | 0.31 | 0.00 | 0.54 | 0.08 |
| | | pred=target | 0.18 | 0.44 | 0.23 | 0.28 | 0.19 | 0.50 | 0.31 | 0.00 | 0.55 | 0.08 |
| | SE | all | 0.67 | 0.68 | 0.64 | 0.74 | 0.71 | 0.62 | 0.54 | 0.77 | 0.66 | 0.67 |
| | | pred=target | 0.68 | 0.69 | 0.64 | 0.75 | 0.72 | 0.62 | 0.54 | 0.78 | 0.67 | 0.67 |
| rings-max | NE | all | 0.28 | 0.26 | 0.34 | 0.23 | 0.19 | 0.55 | 0.35 | 0.00 | 0.52 | 0.18 |
| | | pred=target | 0.28 | 0.27 | 0.37 | 0.23 | 0.22 | 0.55 | 0.35 | 0.00 | 0.50 | 0.17 |
| | SE | all | 0.61 | 0.63 | 0.62 | 0.63 | 0.59 | 0.60 | 0.54 | 0.66 | 0.62 | 0.59 |
| | | pred=target | 0.60 | 0.64 | 0.61 | 0.64 | 0.59 | 0.60 | 0.54 | 0.66 | 0.58 | 0.59 |

## A.5 Showcasing that models do not overfit to the chemical space

To test if our models learned to identify chemical patterns defining the data class, we conducted probing experiments by changing important ground-truth atoms to carbons (excluding ring tasks). The results in Table 8 show a dramatic drop in F1 scores (calculated with the same labels but with probed structure), confirming that the model learned to identify underlying ground truth patterns as alternating the molecule with carbon removed information required to perform correct prediction.

To further validate this, we conducted an additional OOD evaluation using molecules from the ZINC database, which represents a different distribution of commercially available compounds compared to our ChEMBL-based training data. The strong performance on this external dataset demonstrates genuine cross-dataset generalization (see Table 9).

We take it a step further by generating synthetic OOD datasets with randomly created atom combinations, which provide the most rigorous generalization test. These molecules follow entirely different distributional properties while maintaining the target pattern recognition task structure. The first dataset, called "OOD$_f$," includes random atom combinations based on the empirical distribution of atom frequencies in organic molecules (e.g., carbon atoms make up 75% of heavy atoms). The last dataset, called "OOD," samples atoms with equal probability, resulting in completely invalid molecules, of which 50% contain the pattern of interest. In both cases, our GNN recognizes these patterns almost perfectly despite the extremely OOD samples. An example of a generated molecule with one halogen atom is provided here: CC1C(SOONOCS)OSC2(SOOOS)SOOSSC3(SSOOSNOC(S)(CN)N(OO)N3N(SO)C1(C)N)C2(Cl)OSN

Table 8: Model F1 Scores for Original and Substituted Molecules

| Task | Original F1 | Substituted to Carbon F1 |
|------|-------------|--------------------------|
| Benzene (B) | 99.80±0.24 | 0.00±0.00 |
| Pyridine (P) | 99.91±0.11 | 0.86±1.57 |
| Xanthine (X) | 99.97±0.02 | 0.18±0.20 |
| Indole (indole) | 99.33±0.13 | 8.55±2.18 |
| PAINS (PAINS) | 94.02±1.00 | 39.63±3.91 |

Table 9: Out-of-Distribution Performance

| Task | ZINC | $OOD_f$ | OOD |
|------|------|---------|-----|
| X | 100% | 99% | 99% |
| B | 100% | 100% | 100% |
| P | 99% | 100% | 99% |
| indole | 98% | 98% | 100% |
| PAINS | 84% | - | - |
| rings-count | 99% | - | - |
| rings-max | 93% | - | - |

## A.6 JUSTIFICATION OF THE DATASET SIZE

Our design philosophy for this benchmark, intentionally balances the simplicity-complexity trade-off. Our aim was to create a dataset that is:

- **Sufficiently Compact for Rapid Iteration:** A smaller dataset allows researchers to quickly develop, test, and iterate on new models and XAI methods without extensive computational resources or long training times. This accelerates the research cycle.
- **Complex Enough to Reflect Real-World Challenges:** While not encompassing the entirety of chemical space, the seven carefully selected tasks represent a diverse set of common challenges in AI for small molecules. These tasks, ranging from classification to regression-like predictions (e.g., ring count), cover fundamental chemical principles and allow for a robust evaluation of XAI methods' ability to identify relevant features.

To further justify our chosen dataset size, we conducted an analysis evaluating model performance across different dataset scales, ranging from 5,000 to 100,000 samples. The results (see Table 10), presented in the table below, demonstrate that for most tasks, performance largely stabilizes with 50,000 samples.

Table 10: Model Performance across Different Dataset Scales

| Task | 5K Samples | 10K Samples | 25K Samples | 50K Samples | 100K Samples |
|------|-----------|-------------|-------------|-------------|--------------|
| B | 99.92 | 99.92 | 99.96 | 99.96 | 99.94 |
| P | 99.97 | 99.99 | 99.98 | 99.98 | 100.00 |
| X | 99.77 | 99.85 | 99.94 | 99.94 | 99.96 |
| indole | 94.11 | 95.54 | 97.65 | 98.32 | 98.24 |
| PAINS | 82.79 | 85.81 | 91.91 | 92.90 | 93.20 |
| rings-count | 86.86 | 93.14 | 99.64 | 99.62 | 99.94 |
| Rings-max | 92.12 | 91.96 | 92.54 | 92.98 | 92.89 |

The core assumption when creating our dataset was to include simple and moderately complex patterns to achieve two key objectives: (1) to ensure that any GNN can easily learn these patterns, putting focus on the evaluation of XAI methods, and (2) to provide high-quality ground-truth annotations for substructures that directly relate to the instance label. Although simple, these tasks remain relevant for real-world scenarios. For example, the same set of PAINS rules is used in drug discovery to

determine if a molecule may interfere with the assay, helping reduce screening costs by filtering out questionable candidates. In practice, evaluating XAI methods with experimental data is challenging due to significant noise and complex interactions between chemical groups, such as intramolecular interactions or steric effects. Therefore, initial evaluation on simple, well-defined tasks is essential for developing effective explainers, especially since many popular methods fail on our straightforward benchmarks.

## A.7 CHOICE OF F1 FOR CLASSIFICATION EVALUATION

We use weighted F1 as our primary metric because it accounts for class imbalance by aggregating per-class performance proportionally to class frequency. For completeness, in Table 11 report micro- and macro-F1 to capture overall performance and equal-per-class performance, respectively.

Table 11: Comparison of F1 metric variants for GIN classification performance.

| Task | weighted-F1 | micro-F1 | macro-F1 |
|---|---|---|---|
| B | 99.96±0.05 | 99.95±0.04 | 99.50±0.04 |
| P | 99.98±0.03 | 99.97±0.03 | 99.94±0.07 |
| X | 99.94±0.02 | 99.94±0.02 | 99.94±0.02 |
| indole | 98.32±0.36 | 98.31±0.36 | 98.19±0.38 |
| PAINS | 92.90±0.54 | 92.91±0.51 | 91.94±0.63 |
| rings-count | 99.62±0.21 | 99.62±0.21 | 99.54±0.25 |
| rings-max | 92.98±0.84 | 92.36±1.46 | 70.01±2.22 |

## A.8 EXAMPLES OF EXPLANATIONS

Figures 10–16 present examples of node explanations for the GIN classifier for each task using the evaluated explainers. The colors in the null explanations are scaled to highlight outlier scores based on the IQR method applied in our evaluation. Specifically, scores below $Q1 - 1.5 \times IQR$ or above $Q3 + 1.5 \times IQR$ are considered outliers, where Q1 and Q3 represent the 25th and 75th percentiles, respectively, and $IQR = Q3 - Q1$.

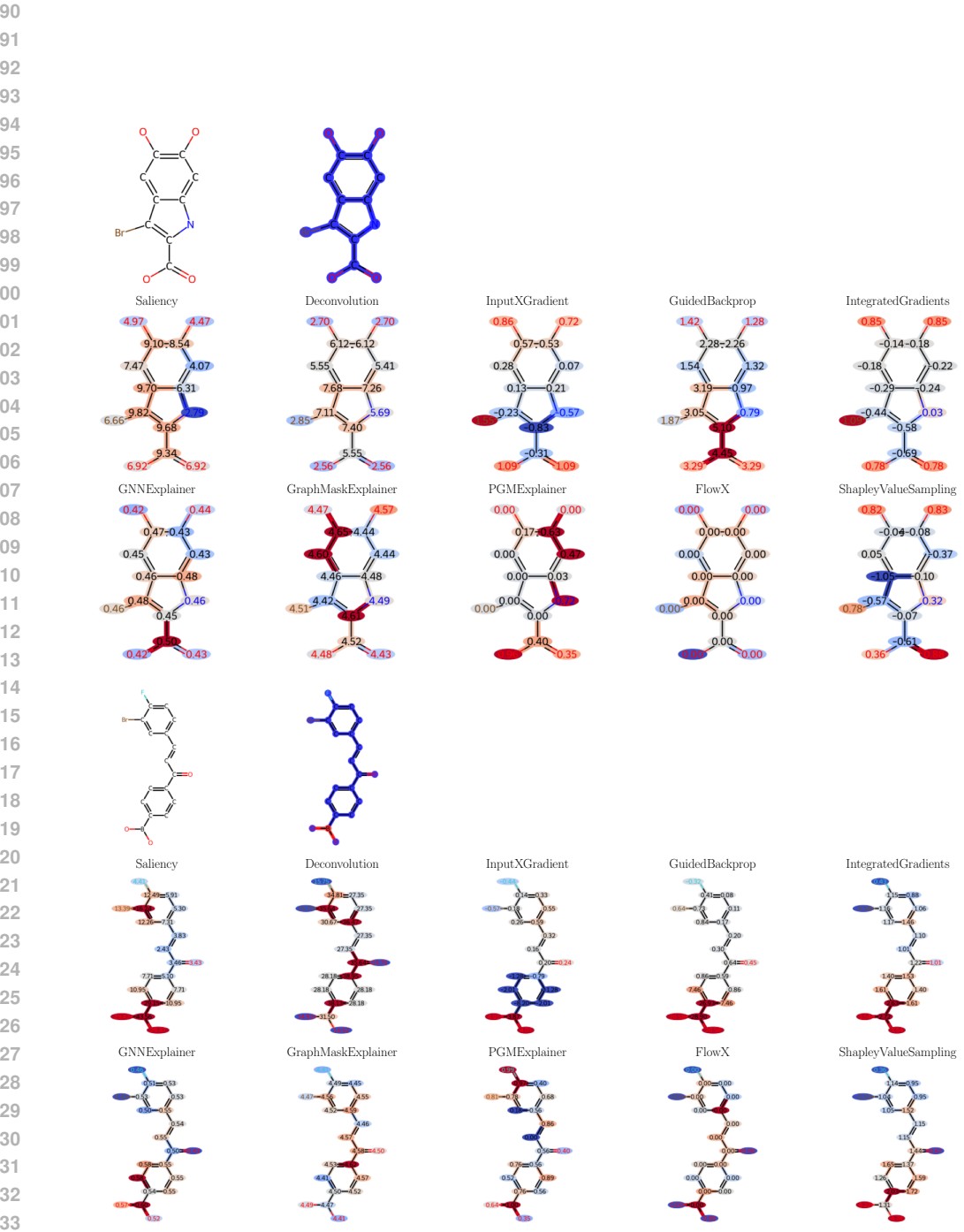

Figure 10: Node-level explanation examples on graphs from different classes in the **B** task, using the GIN model and different explanation methods.

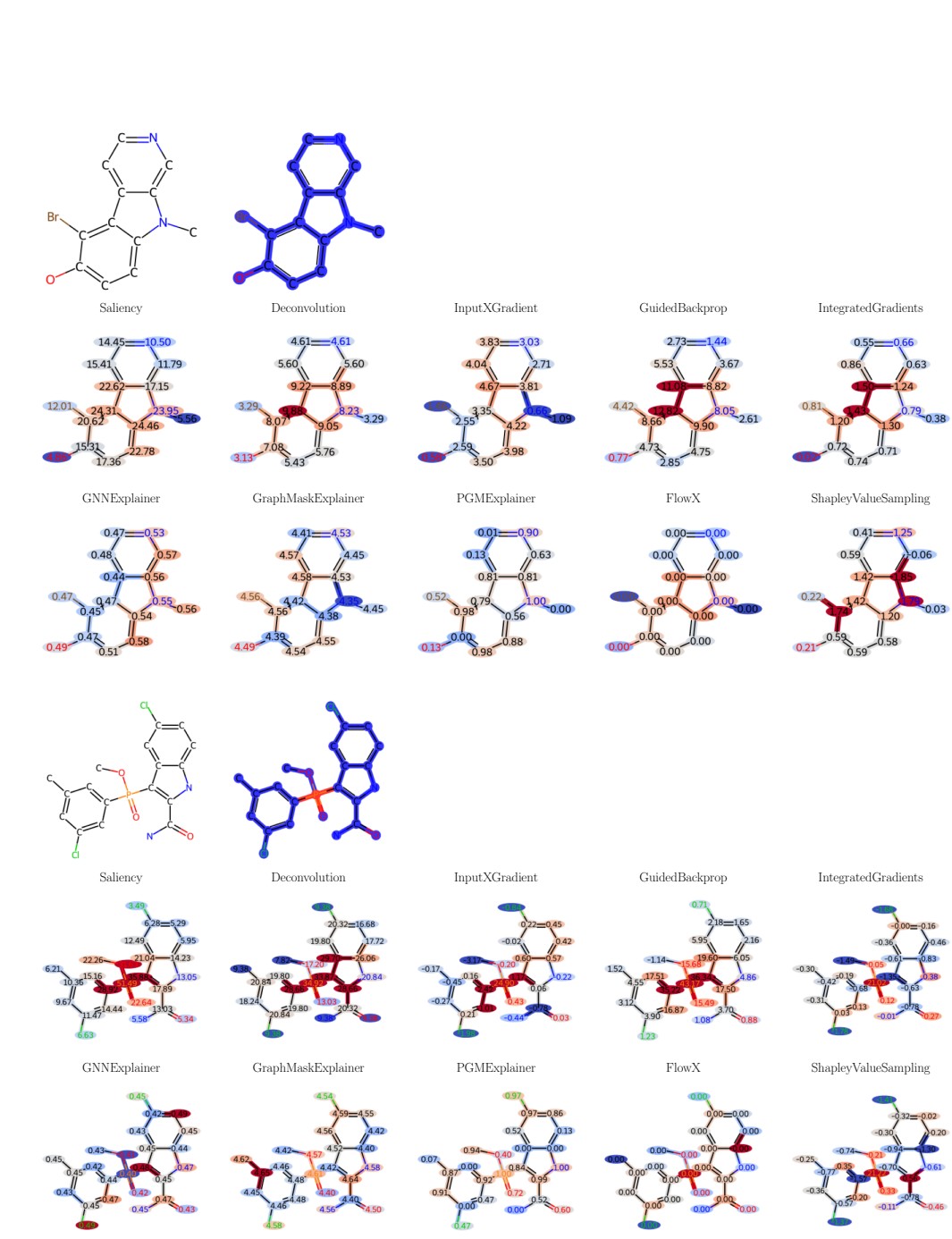

Figure 11: Node-level explanation examples on graphs from different classes in the **P** task, using the GIN model and different explanation methods.

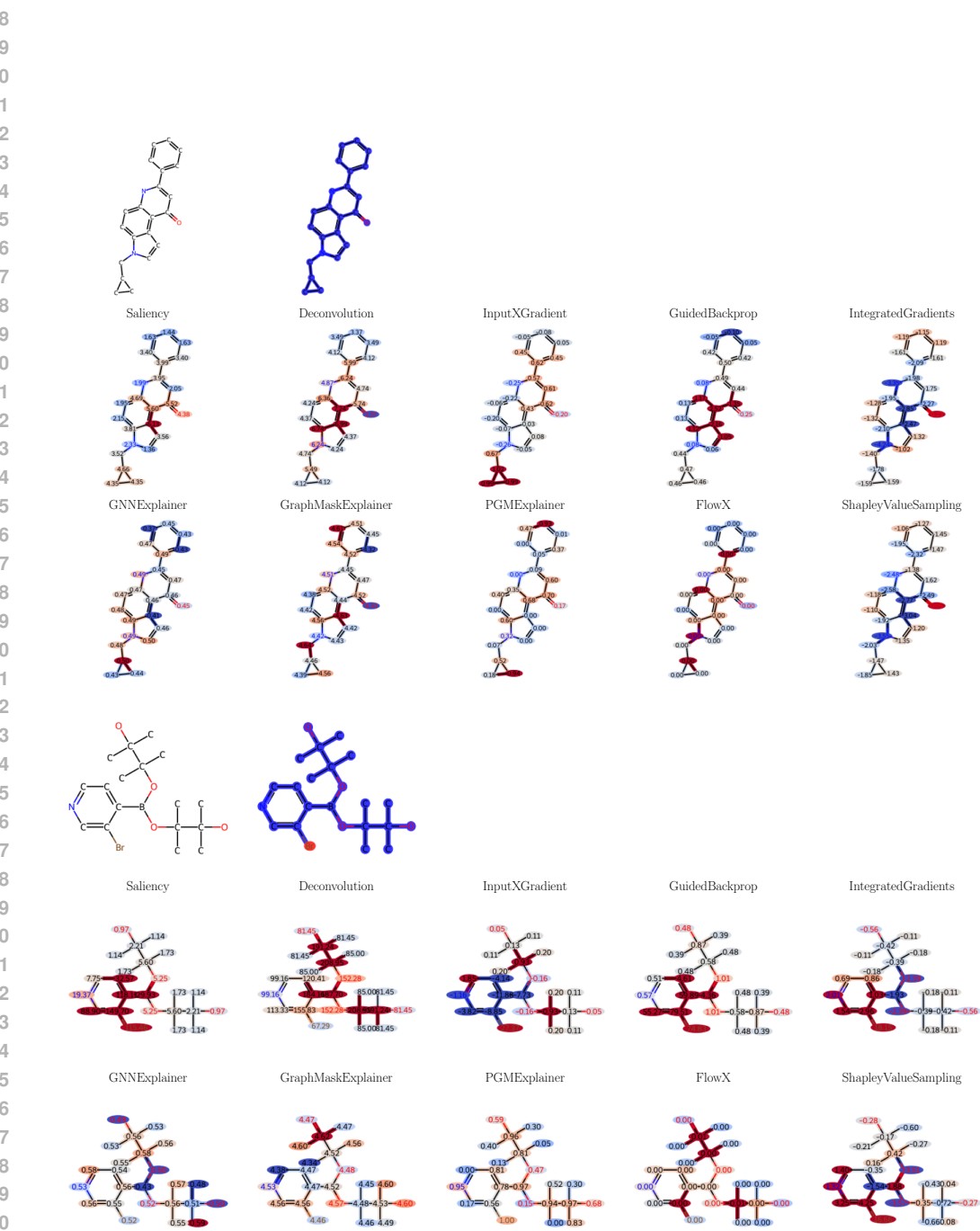

Figure 12: Node-level explanation examples on graphs from different classes in the **X** task, using the GIN model and different explanation methods.

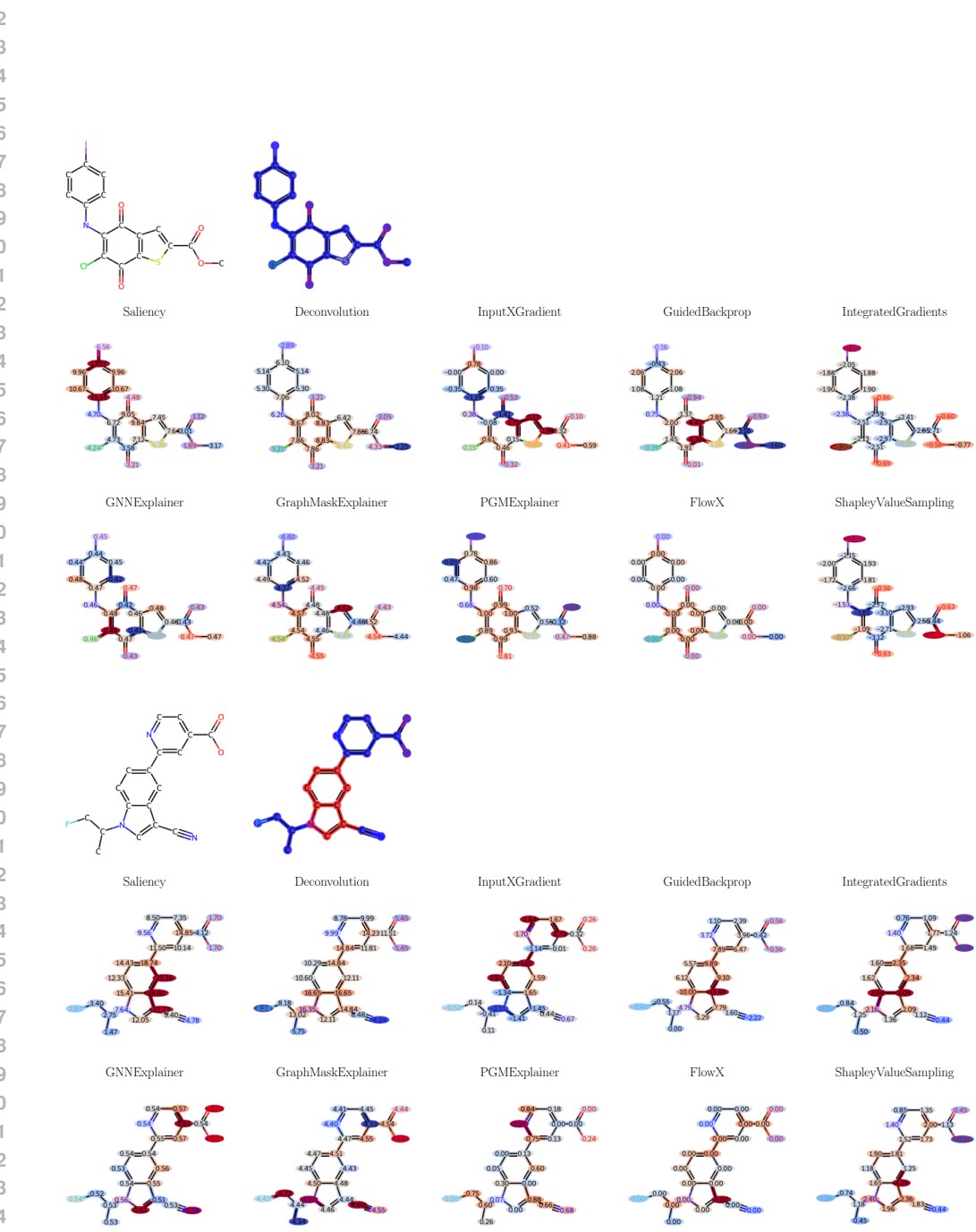

Figure 13: Node-level explanation examples on graphs from different classes in the **indole** task, using the GIN model and different explanation methods.

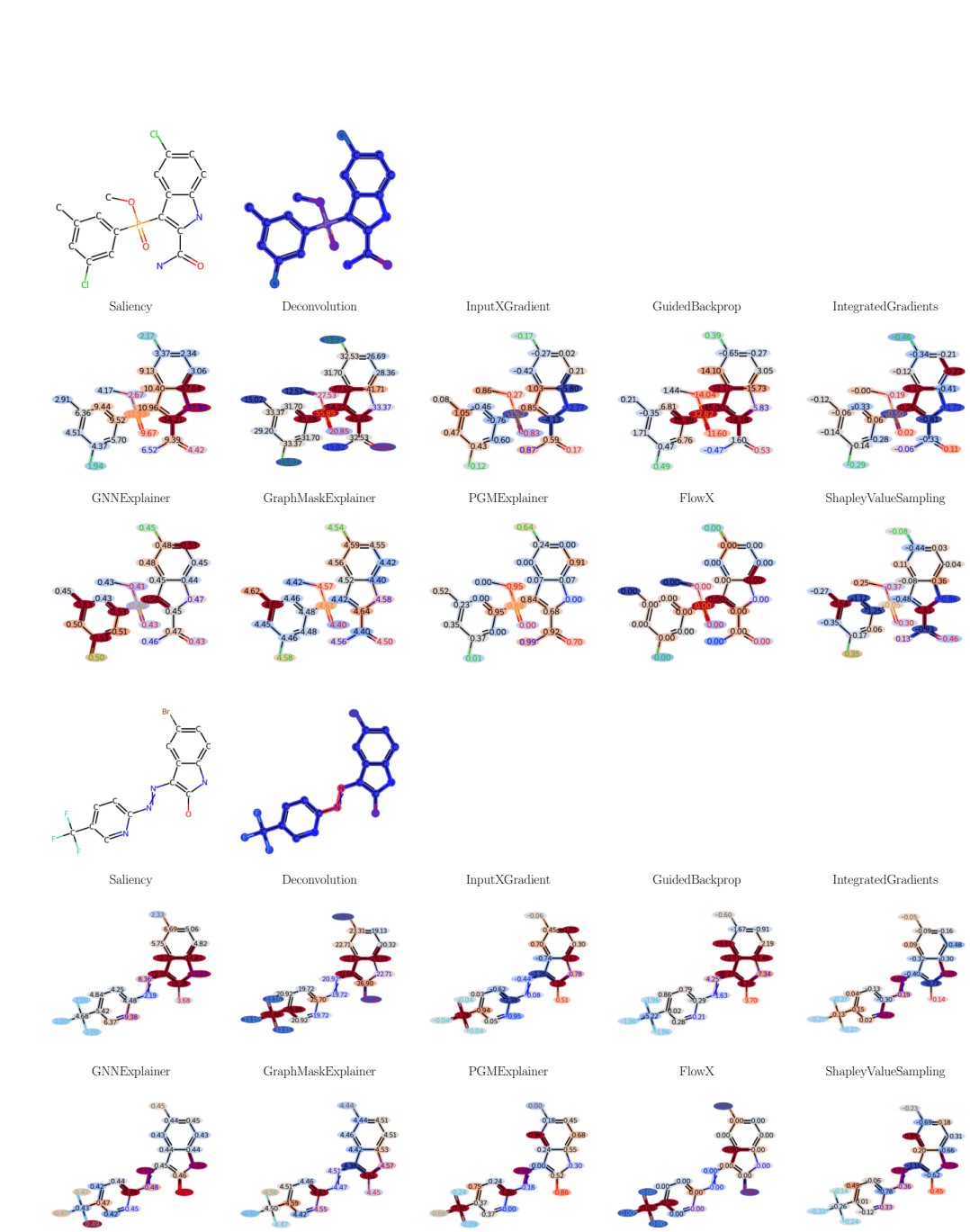

Figure 14: Node-level explanation examples on graphs from different classes in the **PAINS** task, using the GIN model and different explanation methods.

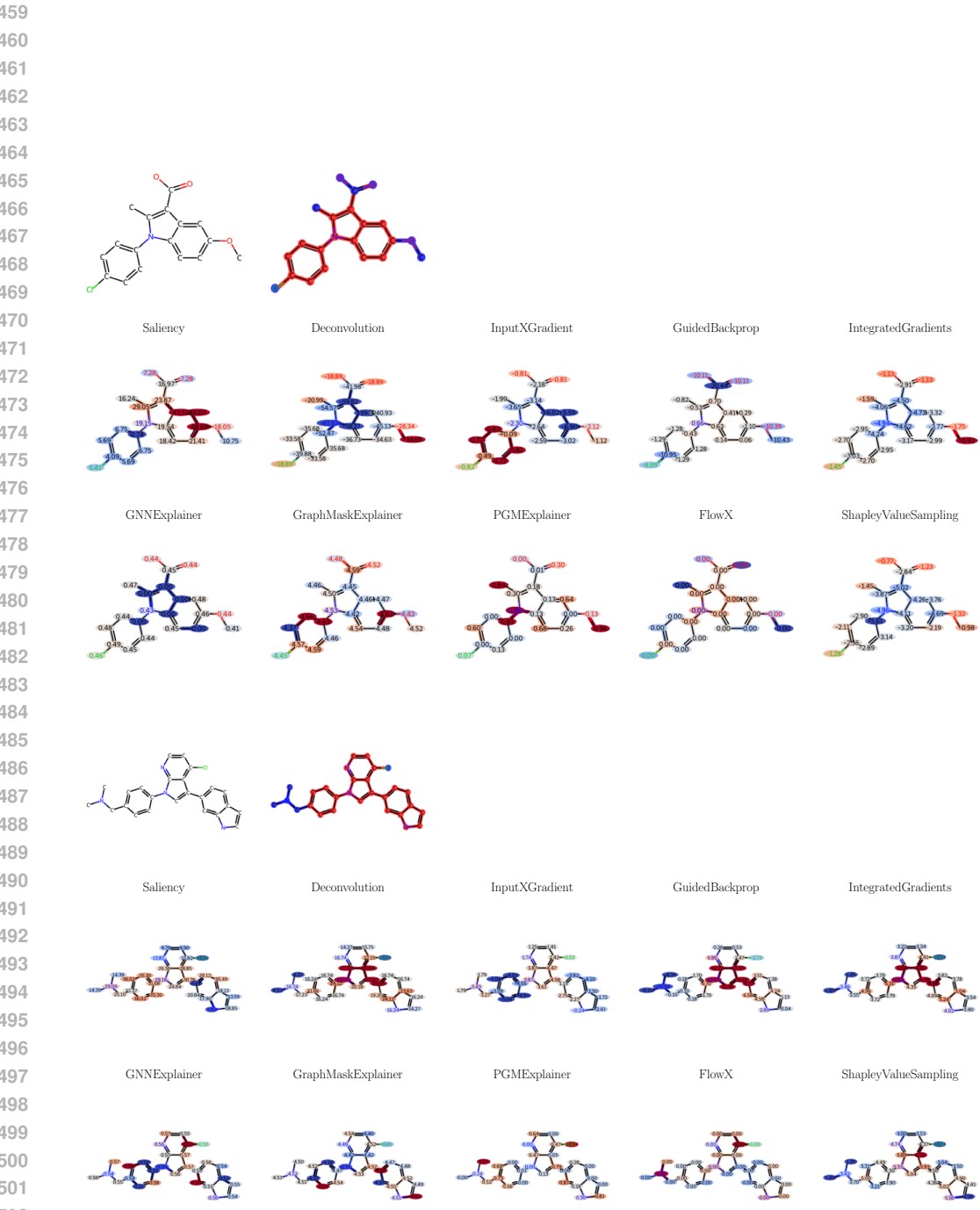

Figure 15: Node-level explanation examples on graphs from different classes in the **rings-count** task, using the GIN model and different explanation methods.

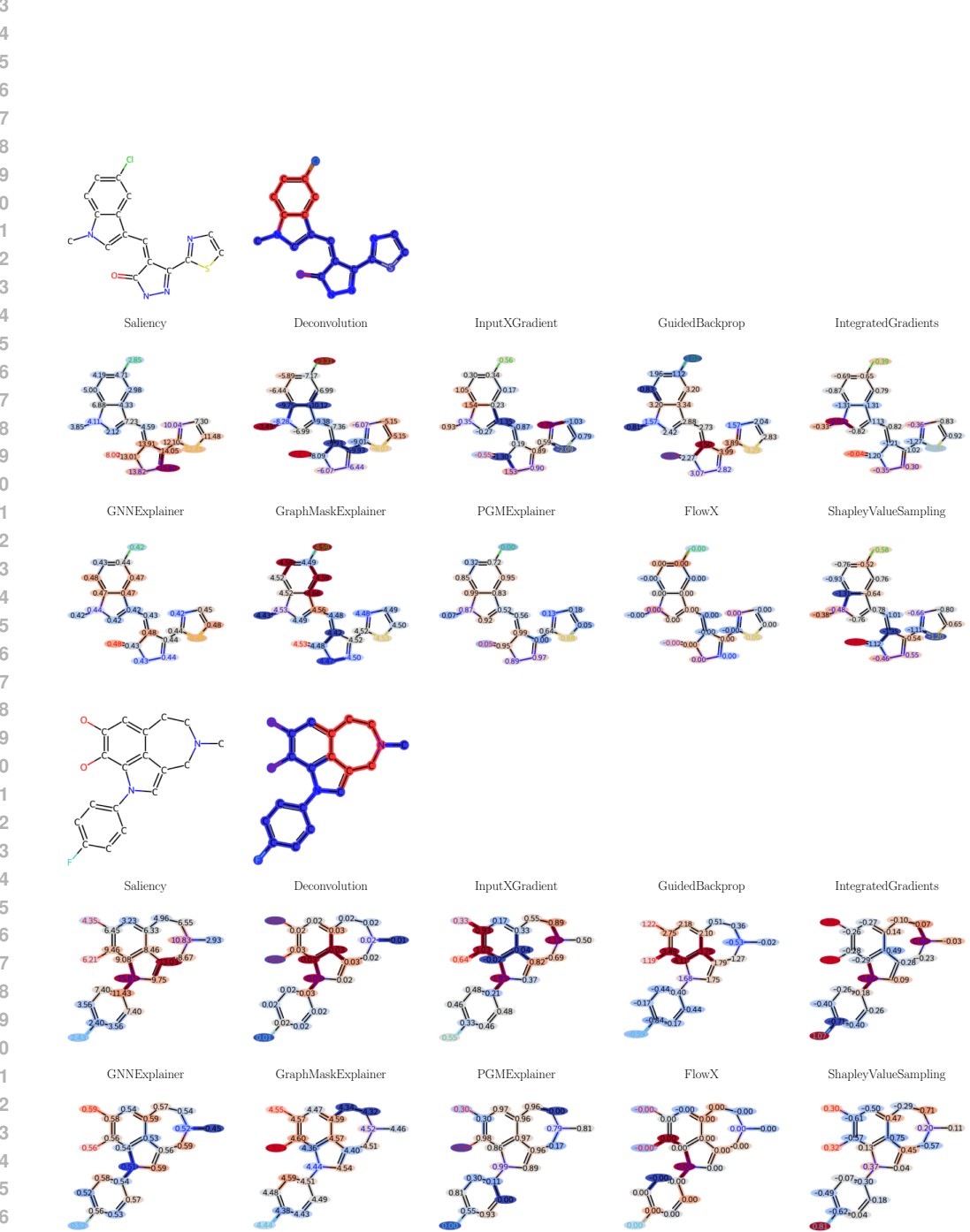

Figure 16: Node-level explanation examples on graphs from different classes in the **rings-max** task, using the GIN model and different explanation methods.

