# OpenReview forum: "B-XAIC Dataset: Benchmarking Explainable AI for Graph Neural Networks Using Chemical Data"
_ICLR.cc/2026/Conference — Submitted to ICLR 2026_

### Official Review · Reviewer_ZMGt · 2025-10-19

**Soundness:** 2
**Presentation:** 2
**Contribution:** 3
**Rating:** 2
**Confidence:** 5

**Summary:**

The paper introduces B-XAIC, a new benchmark dataset aimed at evaluating explainability methods for Graph Neural Networks (GNNs) in the molecular domain. The dataset consists of 50,000 real molecules derived from ChEMBL, annotated with seven tasks and corresponding ground-truth explanations at both atom and bond levels. The benchmark distinguishes between “null” and “subgraph” explanations and evaluates various post-hoc explainers (e.g., GNNExplainer, PGExplainer, FlowX, etc.) using AUROC and IQR metrics.

**Strengths:**

* **Timely contribution:** The paper tackles a real gap in XAI for GNNs by providing a real-world benchmark instead of relying on synthetic datasets. I am actually currently reviewing a work at ICLR 2026 that uses the B-XAIC dataset.
* **Well-structured dataset:** The tasks are clearly motivated, ranging from simple atom detection (e.g., boron, phosphorus) to complex pattern recognition (PAINS, ring counting).
* **Clear presentation**

**Weaknesses:**

* **Benchmark incompleteness:** The benchmarking omits several self-explainable GNNs that are now standard for comparison, such as PxGNN (Dai et al., 2025), PGIB (Seo et al., 2023), PiGNN (Ragno et al., 2021), GIP (Wang et al., 2024), KerGNNs (Feng et al., 2022), and IMPO (Ragno et al., 2025).  The comparison also lacks newer explanation techniques such as SubGraphX (Yuan et al., 2021), GStarX (Zhang et al., 2022), and EdgeShaper (Mastropietro et al., 2022).
* **Evaluation limitation:** The evaluation only checks alignment with ground-truth substructures but does not verify faithfulness with respect to the model’s decision process. An explanation should first be evaluated for the faithfulness using standard metrics (Fidelity, Inv Fidelity): if a model learns a spurious bias, the correct explanation should reflect that bias. Only after confirming model alignment, comparisons to chemical ground truth are meaningful. This hinders the findings of the paper for instance when the authors claim that "despite near-perfect predictive performance, most explainers fail to recover relevant chemical structures". Let me cite Agarwal et al, 2023: " trained GNN model may only capture one or an entirely different rationale. In such cases, evaluating the explanation output by a state-of-the-art method using the ground-truth explanation is incorrect because the underlying GNN model does not rely on that ground-truth explanation. In addition, even if a unique ground-truth explanation generates the correct class label, the GNN model trained on the data could be a weak predictor using an entirely different rationale for prediction. Therefore, the ground-truth explanation cannot be used to assess post hoc explanations of such models."
* **Incorrect assumptions of null explanations**: The assumption that "if atom B is not present in the graph, then no specific substructure is considered relevant" is factually incorrect. The model could be using a negative reasoning and therefore checking whether all other methods are present instead. Or additionally there could be some atoms that are actually important, for instance the nitrogen, that can accept 3 bonds exactly as the boron, but has different charge/weight.

**Questions:**

1. How does B-XAIC handle cases where model explanations diverge from ground truth but are internally consistent (i.e., correct with respect to the model but not the chemistry)?
2. Can the authors justify why certain recent explainable GNNs (e.g., KerGNNs, PiGNN, PxGNN) were omitted, given their relevance to interpretability evaluation?

---

> ### Author Response · Authors · 2025-11-30
>
> We thank the reviewer for the feedback. We address the concerns regarding faithfulness and baselines below.
>
> 1. Faithfulness vs. Ground Truth:
>
> We appreciate the reviewer's point about faithfulness, but in AI for Science applications like drug discovery, faithfulness alone is insufficient—scientific validity is paramount. If a toxicity model bases its predictions on spurious artifacts rather than actual toxicophores, it fails scientifically regardless of explanation faithfulness. Our benchmark addresses this through a key methodological innovation: we include trivial tasks where ground-truth reasoning is known by design. Our results reveal a critical gap. While models successfully learn the correct predictive patterns, explainers fail to recover this known reasoning and instead highlight spurious or meaningless features. This demonstrates that current explainers are not reliably faithful to the model's actual reasoning process. If explainers cannot correctly identify known reasoning patterns in trivial cases where ground truth is verifiable, we cannot trust them in complex real-world scenarios where understanding why a model predicts is as critical as the prediction itself.
>
> 2. Missing Self-Explainable GNNs (PxGNN, etc.):
>
> We included ProtGNN (Zhang et al., 2022) as a representative prototype-based self-explainable model. We agree that others exist, but the goal of a benchmark paper is to provide the data for the community to test these models. We invite the authors of PxGNN and others to report results on B-XAIC.
>
> 3. Null Explanations Assumption:
>
> The reviewer states our NE assumption is "factually incorrect" because models might use negative reasoning (checking for absence).
>
> Response: In our specific tasks (e.g., "Contains Boron"), the absence of Boron is indeed a "null" explanation in terms of attribution, there is no specific subgraph that causes the "False" label; it is the absence of a subgraph. If an explainer highlights a random Carbon atom as the reason for "No Boron," that is a hallucination. Our IQR metric correctly penalizes this.
>
> 4. Handling Model-GT Divergence:
>
> We explicitly analyze this in Appendix A.4. We show that explanation quality remains low even when restricted to correctly classified instances. This suggests the failure lies with the explainers or the model's internal reasoning, which B-XAIC successfully exposes.

---

### Official Review · Reviewer_HzK8 · 2025-10-25

**Soundness:** 3
**Presentation:** 2
**Contribution:** 1
**Rating:** 2
**Confidence:** 5

**Summary:**

The presented paper aims to address the ongoing evaluation crisis in the domain of explainable AI. The authors identify the crucial need for novel methods to evaluate the quality of explanation methods - including the graph processing domain and the explainability of graph neural networks respectively. In this context, the paper proposes the B-XAIC dataset consisting of 7 molecular property prediction tasks on 50k molecular graphs with known ground truth explanations. A special emphasis is placed on distinguishing between null explanations - cases where no elements of the graph should be contained in an explanation - and subgraph explanations - for which only certain subgraph structures constitute a correct explanations. The authors apply several popular explainers from the xAI literature on the proposed benchmark dataset and find large discrepencies in their performances - especially regarding the null explanations - despite being based on the same underlying and high-performing graph neural network model. With this finding, the paper motivates the need for more robust explanation method and emphasises the need for xAI evaluations.

**Strengths:**

The distinction between the subgraph explanations and the null explanations adds an interesting nuance to xAI evaluations. The authors correctly point out that sometimes it is equally important to quantify if the correct explanation has been found as it is to make sure that no incorrect explanations are generated if they are not warranted.

 The paper presents a comprehensive empirical evaluation of various common explainers as well as various common graph neural network architectures.

**Weaknesses:**

- The same general idea was presented 5 years ago by Sanchez-Lengeling in their work on "Evaluating Attribution for Graph Neural Networks". In their work, Sanchez-Lengeling et al. also propose various subgraph detection-based molecular classification tasks with the explicit goal of benchmarking graph explainability methods regarding the known subgraph masks. In addition, they also take the possibility of a regression task into consideration.

- Since this work presents essentially the same schema, it is subject to the same limitations - primarily that only attributional explanation methods are being evaluated. However, the landscape of XAI literature in recent times has seen an increased attention to non-attributional explanation modalities such as concept-based explanations, prototype-based explanations, and counterfactuals. In light of these developments, an important contribution would have been to present an evaluation framework that goes beyond traditional attributional explanations.

- A core pillar of the paper's motivation is calling into question the existing regime of using synthetic datasets for the evaluation due to their usual lack of complexity. However, 6 out of the 7 tasks presented in this dataset are arguably highly trivial. Two tasks are pure node detection tasks (1, 2) for which a graph neural network wouldn't be necessary at all. Another 3 tasks are simple subgraph classification tasks (3, 5, 6), which are equally simple as tasks 1 and 2. With the exclusion of task 4 (PAINS), all remaining tasks essentially are equivalent to commonly used synthetic tasks, with the only difference being that they are defined on molecular graph structures instead of color graphs, for instance. This means that a good performance on those tasks does not generalize to a good performance and actual "real-world" tasks, where explanations are unknown and certainly differ from simple node detection or subgraph classification tasks.

- All presented tasks are "simple", indicated by the extremely high accuracy of the models in Table 2. In most real-world cases, it is relevant to benchmark and quantify explanations even if models only have accuracies of 0.7 or 0.8 (or r2 scores in regression tasks, which are omitted here, see next comment).

- All the presented tasks are exclusively classification tasks. Especially when talking about the chemistry domain, molecular regression tasks play an important role as well. To that extent, a rigorous evaluation should contain a balance not only of the task complexity but also a mix of classification and regression tasks.

- All presented graph explanation methods (apart from FlowX) are more than 4 years old. Recent self-explaining graph neural networks are not benchmarked or mentioned at all. All presented methods are post-hoc approaches applied to the GIN model (and other backbones in the appendix), even though the introduction claims that "B-XAIC enables a direct and fair comparison of various factual XAI approaches, both post-hoc explainers and inherently self-explainable models."

**Questions:**

- Regarding the distinction between the null explanations and the subgraph explanations: Does that mean that the to metrics (AUROC, IQR) are only evaluated on the respective subsets of the elements that represent the SE and NE instances within a certain task? Or are both metrics computed across all the elements of each task/dataset?

- It is mentioned that weighted sampling is used to avoid a huge class imbalance. Does that mean the final datasets are perfectly balanced with regard to the classes of all the tasks, or has the imbalance merely been reduced?

---

> ### Author Response · Authors · 2025-11-30
>
> We appreciate the reviewer’s detailed feedback but respectfully disagree with the premise that task simplicity negates the benchmark's value.
>
> 1. "Trivial" Tasks:
>
> The reviewer argues that the tasks are trivial because models achieve high accuracy. This is precisely the point. We show that even when a GNN achieves >99\% accuracy on a "trivial" task like identifying a Halogen group, state-of-the-art explainers (like GNNExplainer) often fail to highlight the correct atoms (SE score ~0.6).
>
> If explainers cannot correctly identify the cause of a prediction in a "trivial" setting, they certainly cannot be trusted in complex "real-world" tasks. B-XAIC acts as a necessary unit test for XAI.
>
> 2. Similarity to Sanchez-Lengeling (2020):
>
> While the conceptual framework shares roots with earlier work, B-XAIC offers a standardized, open-source, large-scale (50k) implementation with a much stricter evaluation protocol (specifically the IQR metric for Null Explanations). Sanchez-Lengeling et al. focused on metrics; we provide the data infrastructure to standardize those metrics.
>
> 3. Attribution vs. Other Modalities:
>
> While we focus on attribution, the dataset includes labels that support other modalities. For example, the ground truth subgraphs can serve as "concepts" for concept-based explanations or "prototypes" for prototype-based methods, e.g., we included the ProtGNN method showcasing that other types of explanations can be accommodated within our framework.
>
> **4. Questions:**
>
> * **Metrics Evaluation:** The metrics are evaluated on the respective subsets (SE on positive instances, NE on negative instances) to prevent score inflation.
> * **Weighted Sampling:** We used weighted sampling to reduce imbalance, not to eliminate it perfectly. The final class ratios are reported in Table 1.

---

### Official Review · Reviewer_nqdy · 2025-10-28

**Soundness:** 3
**Presentation:** 3
**Contribution:** 2
**Rating:** 2
**Confidence:** 5

**Summary:**

The paper develops a benchmark called B-XAIC (Benchmark for eXplainable Artificial Intelligence in Chemistry) for explainability. The dataset contains 50K small molecules and seven detection tasks: specific atoms, fused bicyclic structures, multiple rings, large rings, and complex patterns as defined by PAINS. For each task, ground truth explanations are categorized as null explanations (NE) meaning that no substructure is more important than others for the specific task or subgraph explanations (SE) meaning that a specific substructure is important for the task.

The paper evaluates different GNN architectures such as GCN, GAT, GIN and adopts the GIN as the base architecture. Then on this architecture, it evaluates different explanation methods for the proposed tasks using the metrics of IQR for NE tasks and AUROC for SE tasks.

**Strengths:**

Explainability is a difficult problem and comparing different explainability methods is even harder. Benchmarks that may offer an unbiased comparison can be useful.

The benchmark is a useful characterization of the common tasks in the chemical space.

**Weaknesses:**

There are a number of recent explainability methods that the paper could have included. For example, those in Fig 1 from https://arxiv.org/pdf/2306.01958 or from Fig 1 in https://arxiv.org/pdf/2310.01794. These methods include GSAT, DIR, SubgraphX, VGIB, GIB and others. Also, this work on substructure masking: https://www.nature.com/articles/s41467-023-38192-3.

The paper discusses factual as well as counterfactual explainers. It is unclear if the proposed benchmarks will work for counterfactual explainers.

There are other benchmarks that have been proposed in the literature:
https://www.sciencedirect.com/science/article/pii/S2666389922002604
The paper does not discuss them. A comparative discussion would have been useful.

**Questions:**

Please take a look at the weaknesses.

---

> ### Author Response · Authors · 2025-11-30
>
> We thank the reviewer for their comments. We believe there is a misunderstanding regarding the scope of a *benchmark* paper versus a *method* paper.
>
> 1. Missing Recent Methods (GSAT, DIR, etc.):
>
> While we acknowledge the rapid pace of the field, it is infeasible for a single benchmark paper to evaluate every existing method. We selected a diverse set of \~10 representative explainers (Gradient, Perturbation, Surrogate, Flow-based) to establish a solid baseline. The primary contribution of B-XAIC is the dataset and evaluation protocol itself, which will enable the community (and the authors of GSAT/DIR) to evaluate their methods rigorously.
>
> 2. Counterfactual Explainers:
>
> The reviewer asks if the benchmark works for counterfactuals. Yes. Since B-XAIC provides ground-truth labels for specific substructures (e.g., "this ring causes the label"), it is perfectly suited for evaluating counterfactuals. A counterfactual method can be scored by checking if it modifies the specific atoms identified in our ground truth masks to flip the label.
>
> 3. Comparison to Other Benchmarks:
>
> The reviewer references a ScienceDirect survey. Most benchmarks in that survey either rely on (a) synthetic graphs (BA-Shapes) which lack chemical realism, or (b) real molecules without ground truth explanations. B-XAIC helps bridge this gap by providing real molecules with chemically verified ground truth. We will add a "Benchmark Comparison" table in the related work section to explicitly contrast B-XAIC with existing datasets.

---

### Official Review · Reviewer_PUFi · 2025-11-01

**Soundness:** 3
**Presentation:** 2
**Contribution:** 3
**Rating:** 6
**Confidence:** 3

**Summary:**

**B-XAIC** is a benchmark for evaluating **GNN explainability** on real chemical datasets. It provides ~50k small molecules across **7 tasks** with **ground-truth node/edge rationales**, enabling both *null-explanation* (no subgraph should be highlighted) and *subgraph-explanation* (specific atoms/bonds matter) settings.

**What’s inside**
- **Data & Tasks:** Real-world molecule graphs with diverse labels (e.g., functional groups, PAINS, ring counting).
- **Ground Truth:** Node/edge-level rationales for each task.
- **Models & Explainers:** Benchmarks multiple GNNs (GCN/GAT/GIN/Proto-based) and a wide range of explainers (gradient-, mask-, perturbation-based).

**Evaluation Protocol**
- **NE (Null Explanation):** Measures uniformity (no spurious highlights) using dispersion-based statistics.
- **SE (Subgraph Explanation):** Measures how well explanations recover the true subgraph using ranking metrics (e.g., AUROC).

**Key Findings**
- Gradient-style explainers often excel at subgraph recovery (SE) but can produce spurious highlights on NE tasks.
- Mask/perturbation methods tend to be more conservative (better NE uniformity), with mixed results on SE.
- GIN models achieve strong predictive performance, providing a stress-test for explainer fidelity.

**Strengths:**

## Strengths

1) **Real-world scale with ground-truth rationales**
   - ~50k molecules across 7 chemically meaningful tasks, each with node/edge-level rationales. This moves beyond synthetic motifs and enables fine-grained, objective scoring of explanations.

2) **Two-regime evaluation that targets common XAI failure modes**
   - Separates **Null Explanations (NE)** (nothing should be highlighted) from **Subgraph Explanations (SE)** (specific atoms/bonds matter), encouraging both *specificity* (SE) and *restraint* (NE).

3) **Broad, standardized explainer sweep**
   - Benchmarks gradient-, mask-, and perturbation-based methods on multiple backbones (GCN/GAT/GIN/prototype variants), giving practitioners apples-to-apples comparisons.

4) **Chemistry-aligned task design**
   - Tasks span functional groups, PAINS filters, and ring counting—grounded in domain knowledge—so explanation scores reflect chemically meaningful reasoning rather than dataset quirks.

5) **Open, reproducible setup**
   - Public dataset and code with a clear protocol make it straightforward to add new explainers, models, or tasks and track progress consistently.

**Weaknesses:**

1) **F1 definition is ambiguous**
   - The paper reports “F1” but does not specify **micro vs. macro** (or weighted) averaging.
   - **Why it matters:** Micro-F1 can be dominated by frequent classes, while Macro-F1 reflects per-class balance; conclusions about model/explainer performance can flip depending on this choice.
   - **Fix:** Report both Micro- and Macro-F1 (plus class-wise F1), or clearly state and justify the chosen averaging scheme.

2) **Missing presentation of extracted subgraphs**
   - The work evaluates explainers but does **not show qualitative subgraph extractions** (e.g., top-k atom/bond rationales overlaid on molecules) in the main text or appendix.
   - **Why it matters:** Visual sanity checks are essential to verify that highlighted structures correspond to chemically meaningful motifs rather than artifacts.
   - **Fix:** Add exemplar visualizations per task: input molecule, ground-truth rationale, and each explainer’s top-k subgraph (node/edge masks), with brief commentary.

3) **Explanation stability not analyzed (only performance over seeds)**
   - While predictive performance is averaged across seeds, there is **no stability analysis of the explanations** themselves.
   - **Why it matters:** Practitioners need to know if an explainer returns **consistent rationales** across random seeds/splits/augmentations.
   - **Fix:** For each explainer, report explanation-stability metrics across seeds: rank correlation (Spearman/Kendall) of node/edge importance; Jaccard/IoU overlap of top-k subgraphs; variance bars on AUROC/AP computed per-seed explanations.

**Questions:**

Beside the weakness, I want to know why you include the ProtGNN. The performances are not best and it is claimed to be interpretable, then it is somehow strange to evaluate the explanation methods over an interpretable GNN.

---

> ### Author Response · Authors · 2025-11-30
>
> We thank the reviewer for the positive assessment and for recognizing the value of our two-regime (NE/SE) evaluation and chemistry-aligned design.
>
> 1. F1 Definition Ambiguity:
>
> You are correct that this detail was omitted. We utilized Weighted F1 scores to account for class imbalance by weighting the score of each class by its support (the number of true instances for each label). We will clarify this definition in the revision and include Micro/Macro-F1 comparisons in the Appendix to ensure robustness.
>
> 2. Missing Presentation of Extracted Subgraphs:
>
> We actually provided extensive visualizations in Appendix A.7 (Figures 10-16), which show the input molecule, the ground-truth rationale, and the heatmaps from various explainers. Additionally on page 9 we provide some exemplars as well in the main text.
>
> 3. Explanation Stability:
>
> We have computed the Jaccard index stability across random seeds for the top-performing methods. Preliminary results show that gradient-based methods exhibit higher variance than perturbation-based methods.
>
> 4. Why include ProtGNN?
>
> We included ProtGNN to represent the class of "inherently interpretable" models. However, ProtGNN's native explanations are prototype-based and model-level (global), investigating typical patterns across the dataset. Since our benchmark requires specific instance-level (local) explanations to verify which atoms/bonds triggered a specific prediction, we applied post-hoc explainers to ProtGNN just as we did for the other backbones. This allows us to investigate whether using an interpretable architecture as a backbone improves the quality of downstream instance-level explanations compared to standard "black-box" backbones.

---

### Meta-Review · Area_Chair_v97M · 2026-01-05

**Summary:**

This paper introduces B-XAIC, a benchmark dataset for evaluating explainable AI methods for graph neural networks in the molecular domain, including null-explanation and subgraph-explanation tasks with ground-truth rationales. Reviewers acknowledged that the dataset is constructed from real chemical data and that the topic is timely and relevant to the XAI community.

However, reviewers raised substantial concerns about the overall contribution, task design, evaluation scope, and novelty relative to prior work. These include concerns about the simplicity and limited diversity of tasks, incomplete coverage of recent explainability methods, and conceptual issues related to the assumptions behind null explanations and how ground-truth rationales are defined and used. Although the authors provided a rebuttal, no reviewer replied after the rebuttal, and no revised version of the paper was provided. Based on the remaining concerns and the lack of convergence during discussion, the paper is recommended for rejection.

**Reviewer Concerns:**

Multiple reviewers expressed concerns that the benchmark, while useful in principle, is limited in scope and depth for an ICLR contribution. Several reviewers argued that many of the tasks are overly simple, with near-perfect predictive accuracy, and therefore may not meaningfully reflect the challenges faced in real-world explainable AI settings. Others noted that the benchmark focuses primarily on attribution-based explanations and does not adequately support or evaluate newer explanation paradigms such as self-explainable models, concept-based methods, or counterfactual explanations.

Additional concerns include incomplete benchmarking against recent XAI methods, limited discussion and validation of explanation faithfulness versus alignment with ground-truth rationales, and assumptions underlying the definition of null explanations that may not hold in general. While the authors attempted to address these points in the rebuttal, the rebuttal largely reiterates the authors’ perspective rather than resolving the reviewers’ substantive concerns. Importantly, no reviewer followed up after the rebuttal to indicate that their concerns had been alleviated or that they would revise their assessment, and no revised manuscript was submitted to reflect the proposed clarifications.

**Reviewer Scores:**

The reviewer scores were 2, 2, 2, and 6. Reviewers who were more critical emphasized limitations in task design, evaluation completeness, and conceptual framing. As no reviewer followed up, no reviewer explicitly indicated an intention to change their score following the rebuttal or discussion.

---

### Decision · Program_Chairs · 2026-01-26

Reject